# Probing RLVR Training Instability through the Lens of Objective-Level Hacking

**Yiming Dong** [1][2]   **Kun Fu** [2]   **Haoyu Li** [3]   **Xinyuan Zhu** [2]   **Yurou Liu** [2]   **Lijing Shao** [4][5]   **Jieping Ye** [2]   **Zheng Wang** [2]

## Abstract

Prolonged reinforcement learning with verifiable rewards (RLVR) has been shown to drive continuous improvements in the reasoning capabilities of large language models, but the training is often prone to instabilities, especially in Mixture-of-Experts (MoE) architectures. Training instability severely undermines model capability improvement, yet its underlying causes and mechanisms remain poorly understood. In this work, we introduce a principled framework for understanding RLVR instability through the lens of *objective-level hacking*. Unlike reward hacking, which arises from exploitable verifiers, objective-level hacking emerges from *token-level credit misalignment* and is manifested as system-level spurious signals in the optimization objective. Grounded in our framework, together with extensive experiments on a 30B MoE model, we trace the origin and formalize the mechanism behind a key pathological training dynamic in MoE models: the abnormal growth of the training-inference discrepancy, a phenomenon widely associated with instability but previously lacking a mechanistic explanation. These findings provide a concrete and causal account of the training dynamics underlying instabilities in MoE models, offering guidance for the design of stable RLVR algorithms.

## 1. Introduction

Recent advances in reinforcement learning with verifiable rewards (RLVR) have shown remarkable success in incentivizing the reasoning abilities of large language models (LLMs) (Jaech et al., 2024; Guo et al., 2025), demonstrating substantial potential in mathematics, code generation,

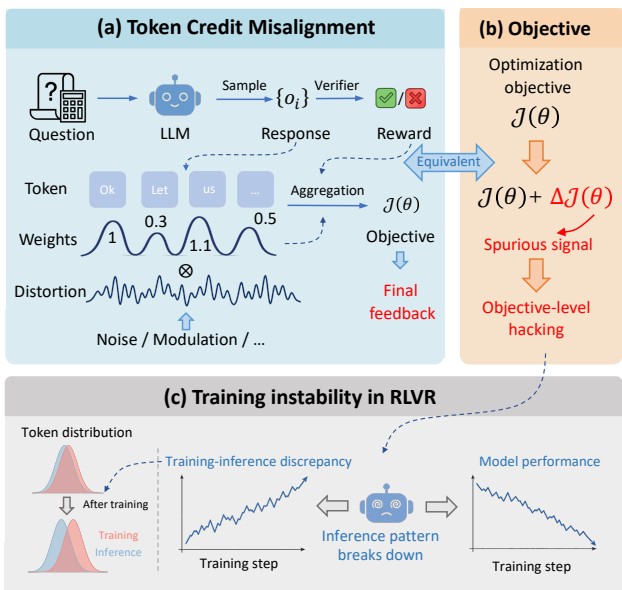

*Figure 1.* **Illustration of the mechanism underlying training instability dynamics in RLVR.** Numerical noise and token-level modulations introduce distortions in token-level weights, which are effectively equivalent to biased perturbations of the optimization objective. Such biases alter the optimization direction toward spurious signals, ultimately resulting in a growing training–inference discrepancy.

and decision-making agents (Shao et al., 2024; Dong et al., 2026; Wu et al., 2025). Compared with supervised fine-tuning, reinforcement learning (RL) methods such as Group Relative Policy Optimization (GRPO) (Shao et al., 2024) exhibit stronger generalization and greater robustness to catastrophic forgetting, demonstrating distinct advantages in long-horizon training (Chu et al., 2025; Jin et al., 2025; Shenfeld et al., 2026). Prior work has shown that prolonged RLVR can drive continuous improvements in the reasoning capabilities of LLMs (He et al., 2025; Liu et al., 2025c). Consequently, maintaining long-term, stable RL training dynamics is crucial for sustained capability gains.

However, training abnormalities in RLVR have been widely reported, and are particularly prevalent in Mixture-of-Experts (MoE) architectures. (Zheng et al., 2025b; Liu et al., 2026; Zhao et al., 2025; Wang et al., 2026). They often manifest as a continue decline in model performance as training progresses, accompanied by anomalies in training dynamics

[1]School of Physics, Peking University [2]Tongyi Lab [3]Alibaba Group [4]Kavli Institute for Astronomy and Astrophysics, Peking University [5]National Astronomical Observatories, Chinese Academy of Sciences. Correspondence to: Zheng Wang <wz388779@alibaba-inc.com>.

*Proceedings of the 43rd International Conference on Machine Learning*, Seoul, South Korea. PMLR 306, 2026. Copyright 2026 by the author(s).

such as token-level entropy and gradient norms.

In response to training instability, numerous variants of the GRPO algorithm and various numerical techniques have been proposed. Such approaches empirically introduce stabilization strategies from different perspectives (Chen et al., 2025; Xue et al., 2026; Zheng et al., 2025b; Zhao et al., 2026; Yao et al., 2025; Zhao et al., 2025; Liu et al., 2025a; He & Lab, 2025), and have, to varying degrees, mitigated or curtailed training instability under specific conditions. However, due to the complexity of the RLVR training process, existing studies on training instability mechanisms remain scarce (Zheng et al., 2025a). In particular, interpretability-focused studies analyzing the anomalous *training dynamics* associated with training instability phenomena are still lacking. This significantly hampers our understanding of the root causes of these instabilities and, consequently, impedes principled algorithm design.

In this work, we introduce a principled framework for understanding RLVR training instability in MoE models through the lens of *objective-level hacking* (Fig. 1). By analogy to *reward hacking*, which arises from vulnerabilities in verifiers that provide spurious signals and thereby disrupt training stability (Cai et al., 2025), objective-level hacking originates from *token-level credit misalignment*. This misalignment induces a system-level spurious signal in the optimization objective, which in turn drives pathological optimization behavior and ultimately destabilizes RLVR training.

Grounded in our framework, we trace the origin and formalize the mechanism behind a key pathological training dynamic in MoE models: the abnormal growth of the training-inference discrepancy (Zheng et al., 2025a; Team et al., 2025). This discrepancy stems from differences in the implementations of inference and training process in RL frameworks, and is widely believed to undermine the stability of RLVR training (Yao et al., 2025). Intriguingly, although such a discrepancy seemingly should be a purely infrastructural artifact, it can nonetheless continue to grow over the course of training (He & Lab, 2025; Zheng et al., 2025a). Within our framework, we attribute this behavior to objective-level hacking arising from token-level credit misalignment, and provide a unified formulation of its underlying mechanism to deepen our understanding of training dynamics associated with instabilities. In summary, our contributions are as follows:

- We offer a new lens for understanding RLVR training instability based on objective-level hacking, providing guidance for the design of RLVR algorithms.

- We construct a unified formulation to understand the abnormal growth of the training-inference discrepancy caused by multiple sources of training instability.

- Through extensive experiments on a 30B MoE model,

we verify that different forms of objective-level hacking are statistically correlated with the anomalous discrepancy growth, which in turn undermines training stability.

## 2. Preliminary

**Group Relative Policy Optimization.** GRPO is adapted from Proximal Policy Optimization (PPO) (Schulman et al., 2017) that replaces value estimation with *group-wise* advantage estimations (Shao et al., 2024). For each question $q \sim \mathcal{D}$, we sample $G$ responses $\{o_i\}_{i=1}^G$ from the old policy $\pi_{\text{train}}(\cdot|\theta_{\text{old}})$, and optimize the current policy $\pi_{\text{train}}(\cdot|\theta)$ via a clipped surrogate objective:

$$\mathcal{J}(\theta)_{\text{GRPO}} = \mathbb{E}_{q\sim\mathcal{D},\,\{o_i\}\sim\pi_{\text{train}}(\cdot|q;\theta_{\text{old}})}\big[\mathcal{L}_{\text{clip}}(\theta)\big]. \quad (1)$$

where

$$\mathcal{L}_{\text{clip}}(\theta) = \frac{1}{G}\sum_{i=1}^{G}\frac{1}{|o_i|}\sum_{t=1}^{|o_i|}\min\Big(r_{i,t}(\theta)\hat{A}_{i,t}, \\ \text{clip}\big(r_{i,t}(\theta),\,1-\varepsilon,\,1+\varepsilon\big)\hat{A}_{i,t}\Big). \quad (2)$$

$r_{i,t} \equiv \pi_{\text{train}}(o_{i,t} \mid q, o_{i,<t};\,\theta)\,/\,\pi_{\text{train}}(o_{i,t} \mid q, o_{i,<t};\,\theta_{\text{old}})$ is the importance weight for the $t$-th token $o_{i,t}$. $\epsilon$ is a hyper-parameter for clip region. The advantage is estimated from group-normalized rewards $\{R_i\}_{i=1}^G$,

$$\hat{A}_{i,t} = \frac{R_i - \text{mean}\big(\{R_i\}_{i=1}^G\big)}{\text{std}\big(\{R_i\}_{i=1}^G\big)}. \quad (3)$$

Following prior works (Hu et al., 2025; Yu et al., 2025), we omit the KL divergence term in the objective.

**Training-inference discrepancy.** In GRPO training, we would expect that all responses $o_i$ should be sampled from $\pi_{\text{train}}(\cdot|\theta_{\text{old}})$ like Eq. (1). However, in current RL frameworks, different implementations are often used for roll-out generation and for model training (Yao et al., 2025; Zhao et al., 2025). This will lead to a mismatch between the token distribution of model used in rollout generation $\pi_{\text{infer}}(\cdot|q;\theta_{\text{old}})$ and model in training $\pi_{\text{train}}(\cdot|q;\theta_{\text{old}})$, so the effective optimization becomes off-policy and the actual optimization objective becomes (Yao et al., 2025),

$$\mathcal{J}'(\theta)_{\text{GRPO}} = \mathbb{E}_{q\sim\mathcal{D},\,\{o_i\}\sim\pi_{\text{infer}}(\cdot|q;\theta_{\text{old}})}\big[\mathcal{L}_{\text{clip}}(\theta)\big]. \quad (4)$$

This can in practice trigger training instability and degrade model performance. Many algorithms therefore draw on importance-sampling principles to correct for this issue and improve training stability (Yao et al., 2025; Liu et al., 2025a; Zhao et al., 2025).

In this work, we use the *variance of importance weights between training and inference at token level* as a fine-grained

indicator of training-inference discrepancy. The importance weight of $\pi_{\text{train}}(\cdot|\theta_{\text{old}})$ and $\pi_{\text{infer}}(\cdot|\theta_{\text{old}})$ reads,

$$\rho_{i,t} = \frac{\pi_{\text{train}}(o_{i,t} \mid q, o_{i,<t}; \theta_{\text{old}})}{\pi_{\text{infer}}(o_{i,t} \mid q, o_{i,<t}; \theta_{\text{old}})} \, . \qquad (5)$$

Ideally, we would expect $\forall o_{i,t}, \; \pi_{\text{infer}} = \pi_{\text{train}}$, which implies $\rho_{i,t} = 1$. However, in practice, implementation inconsistencies prevent this equality from holding exactly. Measuring the variability of $\rho_{i,t}$ therefore provides a way to characterize the magnitude of the distribution mismatch. We define "mismatch" as a global metric to characterize the training-inference discrepancy at a training step, computed as the standard deviation of $\rho_{i,t}$ over all tokens at that step.

## 3. Objective-level hacking

One of the key training dynamics used to monitor instability in MoE models is the *growth* of training-inference discrepancy in RLVR (He & Lab, 2025; Zheng et al., 2025a). It reflects an increasing divergence between the token distributions in training and inference in RL frameworks, despite the fact that model parameters between training and inference are synchronized at every step (Sheng et al., 2025). This key training dynamic is still without a theoretical explanation. We recognize that the *accumulative* nature of this discrepancy suggests that it cannot be attributed to transient stochastic effects at individual training steps, but may instead stem from *an implicit bias in the optimization objective that drives the model parameters toward regions in the parameter space that amplify the discrepancy*. Motivated by this observation, we propose a principled framework to understand this key training instability phenomenon through the lens of the optimization objective analysis.

### 3.1. Implicit bias in optimization objective

We begin by deriving the effects of different sources of training instability on the optimization objective, using the initial training-inference discrepancy as an illustrative example. Ignoring the clipping term for simplicity, the ideal optimization objective of GRPO reads,

$$\mathcal{J}(\theta) = \mathbb{E}_{q\sim\mathcal{D},\{o_i\}\sim\pi_{\text{train}}(\cdot|\theta_{\text{old}})}\left[\frac{1}{G}\sum_{i=1}^{G}\frac{1}{|o_i|}\sum_{t=1}^{|o_i|} r_{i,t}(\theta)\,\hat{A}_{i,t}\right]$$

$$\equiv \mathbb{E}_{\text{train}}\left[\sum_{i,t} X_{i,t}(\theta)\right] \, . \qquad (6)$$

where we denote by $\mathbb{E}_{\text{train}}[\cdot] \equiv \mathbb{E}_{q\sim\mathcal{D}, \{o_i\}\sim\pi_{\text{train}}(\cdot|\theta_{\text{old}})}[\cdot]$ and $\mathbb{E}_{\text{infer}}[\cdot] \equiv \mathbb{E}_{q\sim\mathcal{D}, \{o_i\}\sim\pi_{\text{infer}}(\cdot|\theta_{\text{old}})}[\cdot]$. For notational convenience, we define,

$$X_{i,t}(\theta) \equiv \frac{r_{i,t}(\theta)\,\hat{A}_{i,t}}{G \cdot |o_i|} \, . \qquad (7)$$

The initial training-inference discrepancy acts as a numerical noise perturbing the expected weights of each token, and such perturbation simultaneously leads to modification in the optimization objective. The *effective* optimization objective becomes,

$$\mathcal{J}'(\theta) \; = \; \mathbb{E}_{\text{infer}}\left[\sum_{i,t} X_{i,t}(\theta)\right] \, . \qquad (8)$$

In effect, this implies that our optimization objective can be *interpreted* as containing an additional contribution on top of the original optimization objective,

$$\mathcal{J}'(\theta) = \mathcal{J}(\theta) + \Delta\mathcal{J}(\theta) \, . \qquad (9)$$

After derivations (see Appendix A.1), we obtain,

$$\Delta\mathcal{J}(\theta) \simeq \sum_{i,t} \text{Cov}_{\text{train}}\big(X_{i,t}(\theta), \; \rho_{i,t}^{-1}\big) \, , \qquad (10)$$

where $\text{Cov}_{\text{train}}$ denotes the covariance operator. In addition to explicitly inducing a mismatch in the token distribution, the training-inference discrepancy also exerts an implicit influence on the optimization objective, resulting in the pursuit of a spurious signal $\Delta\mathcal{J}(\theta)$. Such *objective-level* signals may drive the model toward pathological behaviors, such as pursuing *unintended correlations* between $X_{i,t}$ and $\rho_{i,t}^{-1}$. Such unintended and pathological optimization directions can destabilize MoE training, interfering with the model's inherent inference mode, ultimately manifesting as anomalies in the training-inference discrepancy. We refer to this mechanism, by which system-level spurious signals in Eqs. (9) and (10) influence training instability, as *objective-level hacking*.

### 3.2. On the dual effects of token-level weight modulation

Beyond the initial training-inference discrepancy, we point out that the sources of *objective-level hacking* are broader than anticipated. Empirically motivated forms of proactive token weight modulation may similarly introduce implicit bias into the optimization objective.

For example, token-level clipping is originally introduced to prevent excessively large updates and thereby maintain training stability in RLVR (Shao et al., 2024). However, in the context of MoE model training, it could also act as a generalized form of *token-level credit misalignment*, as it effectively leads to a redistribution of the predetermined token weights. The optimization objective with token-level clipping can be written as,

$$\mathcal{J}_{\text{clip}}(\theta) = \mathbb{E}_{\text{train}}\left[\sum_{i,t} \phi_{i,t} \times X_{i,t}(\theta)\right] \, . \qquad (11)$$

where,

$$\phi_{i,t} = \begin{cases} 0, & \text{if } o_{i,t} \in S_{\text{clip}}, \\ 1, & \text{otherwise}. \end{cases} \quad (12)$$

$S_{\text{clip}} = \{o_{i,t} \mid r_{i,t} < 1 - \epsilon_{\text{low}} \vee r_{i,t} > 1 + \epsilon_{\text{high}}\}$ denotes the set of tokens that are subject to clipping. $\mathcal{J}_{\text{clip}}(\theta)$ can also be *interpreted* as (see Appendix A.1),

$$\mathcal{J}_{\text{clip}}(\theta) = \mathcal{J}(\theta) + \Delta_{\text{clip}}\mathcal{J}(\theta), \quad (13)$$

$$\Delta_{\text{clip}}\mathcal{J}(\theta) = \mathbb{E}_{\text{train}}\left[\sum_{i,t} X_{i,t}(\theta)\left(\phi_{i,t} - 1\right)\right]. \quad (14)$$

From the perspective of the optimization objective, this may also lead to unintended perturbations in the model's inference process, such as *unintended correlations* between $X_{i,t}$ and $\phi_{i,t} - 1$, *which were not intended by the original token-level clipping mechanism*.

We present a *unified* formula to characterize the objective-level hacking introduced by token-level credit misalignment. The objective can be interpreted as,

$$\mathcal{J}_{\text{dist}}(\theta) = \mathcal{J}(\theta) + \Delta_{\text{dist}}\mathcal{J}(\theta). \quad (15)$$

Beyond achieving their intended purposes, different forms of token-level weight modulation may also exhibit *dual effects*. Specifically, we will demonstrate through empirical studies that certain forms of $\Delta_{\text{dist}}\mathcal{J}(\theta)$, as objective-level spurious signals, can lead to an increase in discrepancy. Such risks can be exacerbated in MoE models, where inference mechanisms are inherently more complex and sensitive (Zhao et al., 2025; Ma et al., 2025).

## 4. Experiments

We conduct extensive experiments on a 30B MoE model to investigate how objective-level hacking impacts the training-inference discrepancy growth in RLVR. In Section 4.1, we describe the experimental setup. Section 4.2 and 4.3 examine the effects from initial discrepancies and token-level clipping, respectively. In Section 4.4, we design controlled experiments with actively injected token-level weight distortion to further investigate the causality.

### 4.1. Experimental setup

We conduct the experiments based on the Qwen3-30B-A3B model (Yang et al., 2025). In experiments, the model is trained on the DAPO-Math-17k dataset (Yu et al., 2025), with AIME24 used as validation to ensure reproducibility. The experiments are implemented with `verl` framework (Sheng et al., 2025), with vLLM serving as the inference backend (Kwon et al., 2023) and Megatron as the training backend (Shoeybi et al., 2019). The general training

configuration is as follows. At each training step, we sample 128 problems and generate 16 responses per problem, followed by 4 parameter update steps, corresponding to an off-policy training setting. The maximum response length for each response is set to 8K to reduce computational overhead. Unless otherwise stated, for GRPO with token-level clipping (Shao et al., 2024), we use left and right clipping ranges of 0.2; for Group Sequence Policy Optimization (GSPO) (Zheng et al., 2025b) with sequence-level clipping, we set the left and right clipping ranges to 3e-4 and 4e-4, respectively (Zheng et al., 2025b). All experiments are conducted on 4 nodes × 8 A100 GPUs[1].

### 4.2. Hacking from initial training-inference discrepancy

To investigate the impact of the initial training-inference discrepancy on the optimization objective, we construct two comparative settings: a vanilla GRPO setup and a variant that applies truncated importance sampling (TIS) (Yao et al., 2025) to correct the bias of optimization objective. Unlike infrastructure-level interventions (He & Lab, 2025; Ma et al., 2025), TIS does not directly mitigate the token distribution mismatch between training and inference. It instead addresses bias in the optimization objective, making it an ideal setting for analyzing the individual influence of objective-level hacking.

As shown in Fig. 2, TIS markedly reduces the growth rate of the training-inference discrepancy (see Fig. 2c), which indirectly highlights the role of the biased objective in driving the growth of such discrepancies. In addition to mitigating the growth of the discrepancy, TIS slows the decline of token entropy and yields a corresponding improvement in validation scores[2]. However, for MoE training, the training-inference discrepancy is not completely eliminated even when TIS is applied, indicating that other sources may also contribute to its growth.

In Fig. 3, we present the scatter plots of $(\pi_{\text{train}}, \pi_{\text{infer}})$ at the early and later stages of training and compute their Pearson correlation coefficient (PCC). A higher PCC indicates a stronger correlation between the two, while a decrease in PCC signifies an increasing discrepancy. As training progresses, the discrepancy becomes more pronounced, with abnormal clustering observed along the $\pi_{\text{infer}}$ dimension in the later stage. We attribute this clustering to the limited numerical precision in inference mode and discuss its impact on the training dynamics (see Appendix C.1). This perspective further provides evidence for the collapse of the model's inference behavior.

---

[1]The A100 is a non-Hopper architecture GPU with relatively lower numerical precision.

[2]Experiments conducted on dense models reveal similar phenomena related to token entropy (Yao et al., 2025).

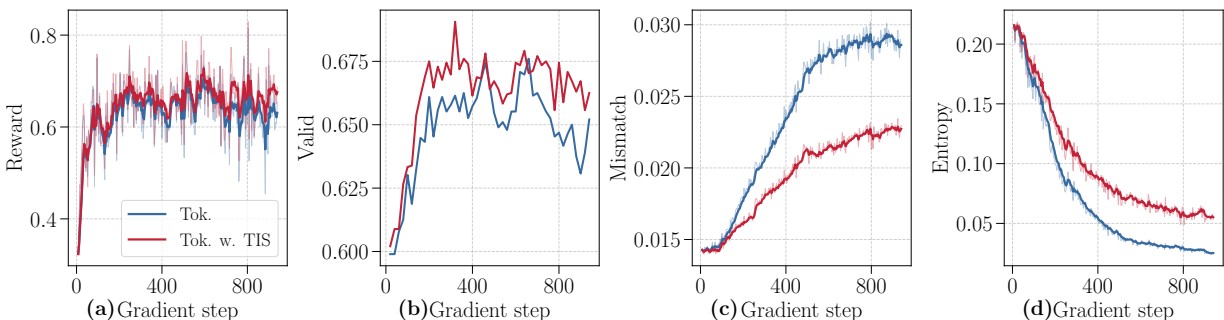

*Figure 2.* **Influence of initial training-inference discrepancy.** The blue curve shows the training dynamics with token-level clipping (Tok.), while the red curve corresponds to the same setting augmented with TIS correction (Tok. w. TIS). From left to right, the four panels show: (a) training reward, (b) validation accuracy, (c) training-inference discrepancy, and (d) token entropy during training.

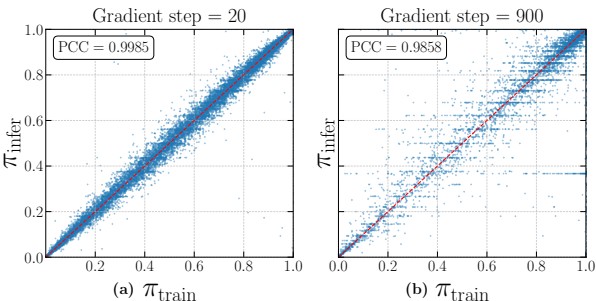

*Figure 3.* **Token probabilities in inference mode vs. training mode.** (a) and (b) subplots are taken from the token-level clipping with TIS correction at gradient steps 20 and 900 in Fig. 2, respectively. The red dashed line marks $y = x$, and the PCC between the two probabilities is annotated in the top-left corner.

### 4.3. Hacking from token-level clipping

In this section, we demonstrate the effects of objective-level hacking induced by token-level clipping. Token-level clipping is commonly believed to stabilize RLVR training. However, we uncover a counterintuitive phenomenon: *stronger token-level clipping actually accelerates the growth of the training-inference discrepancy*.

*Table 1.* **Variation of clip ratio during RLVR training (Fig. 4)**

| Gradient Step | 4 | 200 | 400 |
|---|---|---|---|
| Token (low) | $1.50 \times 10^{-3}$ | $1.15 \times 10^{-3}$ | $1.26 \times 10^{-3}$ |
| Token (middle) | $1.92 \times 10^{-3}$ | $1.81 \times 10^{-3}$ | $2.84 \times 10^{-3}$ |
| Token (high) | $2.37 \times 10^{-3}$ | $2.63 \times 10^{-3}$ | $3.49 \times 10^{-3}$ |
| Seq. | 0.153 | 0.120 | 0.085 |

We design experiments by varying the strength of token-level clipping to observe the corresponding growth of the training-inference discrepancy, and analyze their statistical correlation. We adopt the decoupled clipping strategy from DAPO (Yu et al., 2025) to construct different token-level clipping strengths, fixing the left clipping range at 0.2 and varying the right clipping range over {0.2, 0.24, 0.28}.

As an overall comparison, we additionally present the training dynamics of sequence-clipping-based algorithm GSPO (Zheng et al., 2025b).

As shown in Fig. 4, in MoE training we find that stronger clipping settings (clip ratio see Tab. 1) are associated with a faster increase in the training-inference discrepancy. On the one hand, token-level clipping proves effective in maintaining training stability across a wide range of training scenarios (Shao et al., 2024; Guo et al., 2025; Yu et al., 2025). On the other hand, in certain training regimes, such clipping can introduce biased objective-level signals, potentially leading to implicit effects on training stability. In contrast, under similar settings, sequence-level clipping does not exhibit the same anomalous growth in the training-inference discrepancy, which provides indirect evidence that token-level clipping plays a role in the observed discrepancy growth.

### 4.4. Hacking from explicitly injected token-level weight distortion

To further investigate the *causality* between objective-level spurious signal and training-inference discrepancy, we adopt a *proactive* strategy and design active noise injection experiments to explore its underlying mechanism. Specifically, since sequence-level clipping does not exhibit anomalous growth under our setting, we explore *what types of perturbations can reproduce such growth*. We consider a straightforward form of token-level weight distortion, in which different weights are assigned to tokens with *high* and *low* probabilities. This in turn gives rise to the objective-level spurious signals described in Eq. (15). The initial optimization objective of GSPO reads (Zheng et al., 2025b),

$$\mathcal{J}_{\text{Seq}} = \mathbb{E}_{q \sim \mathcal{D}, \{o_i\} \sim \pi_{\text{train}}(\cdot | \theta_{\text{old}})} \left[ \frac{1}{G} \sum_{i=1}^{G} \frac{1}{|o_i|} \sum_{t=1}^{|o_i|} \right.$$
$$\left. \min \left( s_{i,t}(\theta) \, \hat{A}_{i,t}, \text{clip} \left( s_{i,t}(\theta), 1 - \epsilon, 1 + \epsilon \right) \hat{A}_{i,t} \right) \right].$$
$$(16)$$

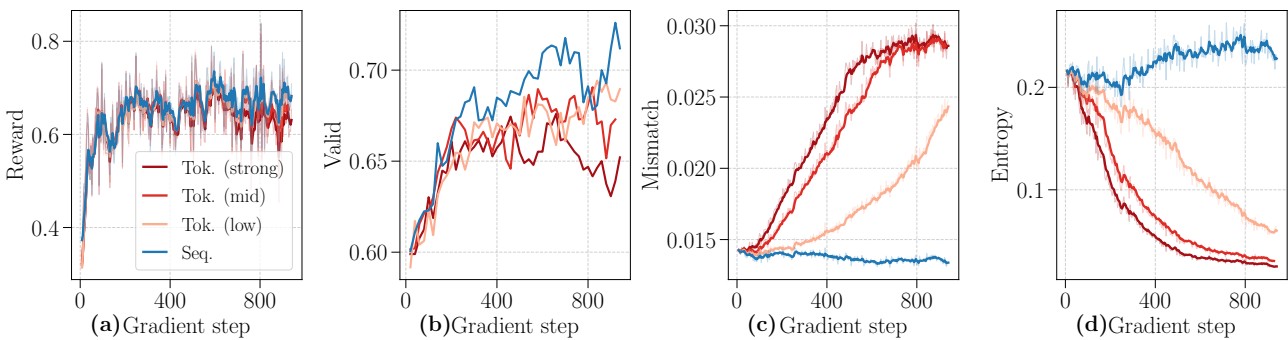

*Figure 4.* **Influence of token-level clipping.** Blue curves correspond to training dynamics with sequence-level clipping (Seq.). Red curves represent token-level clipping with varying strengths, where darker shades indicate stronger clipping and lighter shades represent weaker clipping. In the legend, "strong", "mid", and "low" correspond to right clipping ranges of 0.2, 0.24, and 0.28, respectively.

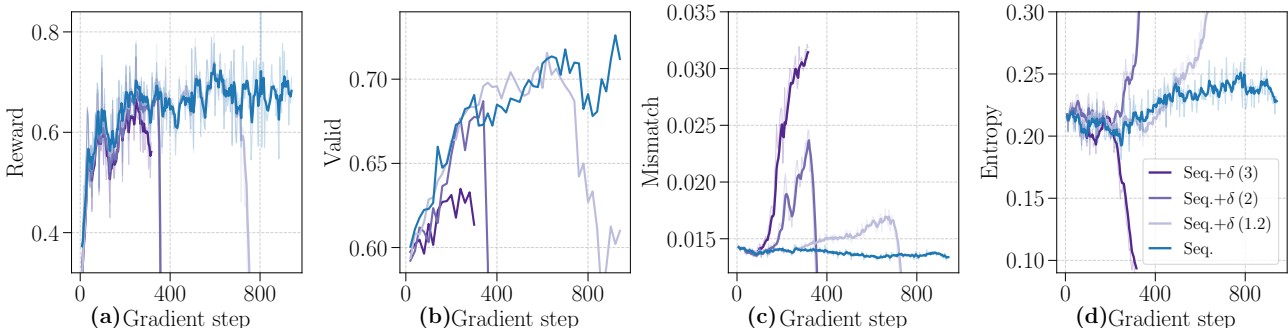

*Figure 5.* **Influence of injected token-level weight distortion.** Blue curves correspond to training dynamics with sequence-level clipping. Purple curves denote the same configuration with different strengths of injected token-level weight distortion, where darker to lighter shades indicate stronger to weaker one, respectively. $\delta$ in the legend indicates the distortion strength defined in Eq. (18).

For covariance, we define,

$$Y_{i,t} \equiv \frac{\min\left(s_{i,t}(\theta)\,\hat{A}_{i,t}, \mathrm{clip}\left(s_{i,t}(\theta), 1-\epsilon, 1+\epsilon\right)\hat{A}_{i,t}\right)}{G \cdot |o_i|}.$$

Under this definition, $\mathcal{J}_{\mathrm{Seq}} = \mathbb{E}_{\mathrm{train}}\left[\sum_{i,t} Y_{i,t}(\theta)\right]$. Based on this, we inject token-level weight distortion into the optimization objective by assigning different credits to high-probability and low-probability tokens. The objective reads,

$$\mathcal{J}_{\mathrm{inj}} = \mathbb{E}_{\mathrm{train}}\left[\sum_{i,t} \varphi_{i,t} \times Y_{i,t}(\theta)\right]. \qquad (17)$$

where,

$$\varphi_{i,t} = \begin{cases} \delta, & \text{if } o_{i,t} \in S_{\mathrm{low\_prob}}, \\ 1, & \text{otherwise}. \end{cases} \qquad (18)$$

where $S_{\mathrm{low\_prob}} = \{o_{i,t} \mid \pi_{\mathrm{train}}(o_{i,t} \mid q, o_{i,<t}; \theta_{\mathrm{old}}) < \pi_{\mathrm{low}}\}$. Intuitively, this modification assigns a weight of $\delta$ to tokens whose probabilities fall below $\pi_{\mathrm{low}}$, while all remaining tokens retain a weight of 1.

We first evaluate the effect of *increasing* the weight assigned to low-probability tokens during MoE training, setting $\pi_{\mathrm{low}} = 0.1$ and varying $\delta \in \{3, 2, 1.2\}$. As shown in

Fig. 5, we observe that injecting such token-level weight distortion reintroduces the similar anomalous increase in the training-inference discrepancy. Moreover, the growth rate of the discrepancy exhibits a statistical correlation with the strength of the token-level weight distortion: *stronger distortions induce faster discrepancy growth*. Along with the growth of the discrepancy, the model's performance exhibits an irreversible degradation. This is accompanied by anomalies in token entropy, which serves as another indicator of the collapse of the model's inference patterns.

In addition to increasing the weight assigned to low-probability tokens, *decreasing* that weight can also reproduce the growth of the training-inference discrepancy (see Appendix B.2). It suggests that the anomaly is fundamentally driven by *weight distortion* rather than by act of increasing weight. Such injected distortion can likewise be described by the similar formula,

$$\mathcal{J}_{\mathrm{inj}}(\theta) = \mathcal{J}_{\mathrm{Seq}}(\theta) + \Delta_{\mathrm{inj}}\mathcal{J}(\theta), \qquad (19)$$

$$\Delta_{\mathrm{inj}}\mathcal{J}(\theta) = \mathbb{E}_{\mathrm{train}}\left[\sum_{i,t} Y_{i,t}(\theta)\left(\varphi_{i,t} - 1\right)\right]. \qquad (20)$$

It also introduces *objective-level* spurious signals that per-

turbs the optimization direction. Notably, even a modest 20% increase in the weight of low-probability tokens is sufficient to trigger the growth of the training-inference discrepancy. This carefully controlled ablation setting provides clearer evidence that *objective-level hacking from token-level weight distortion constitutes an important source of the training-inference discrepancy growth in MoE model training*. This findings provide evidence for the enhanced stability of sequence-level algorithms in MoE training (Zheng et al., 2025b; Liu et al., 2025a), and suggests that algorithms toward MoE models should consistently avoid such dual effects to protect the relatively sensitive inference behavior of MoE models.

## 5. In-depth characterization of the mechanism

In this section, we conduct an in-depth analysis of the characteristics of objective-level hacking mechanisms to better understand their role in driving training instability in MoE models.

### 5.1. Bias instead of variance is the main driver

To better understand the mechanisms of objective-level hacking, we conduct additional experiments to investigate the question: while we observe that *biased* perturbations to the optimization objective can trigger discrepancy growth, *does injecting unbiased, variance-based token-level weight distortion also lead to such growth?*

Similar to Section 4.4, we formulate the optimization objective as,

$$\mathcal{J}_{\text{var}} = \mathbb{E}_{\text{train}} \left[ \sum_{i,t} \xi_{i,t} \times Y_{i,t}(\theta) \right]. \tag{21}$$

where $\xi_{i,t}$ is *independently* sampled from a Gaussian distribution $\mathcal{N}(1, \sigma^2)$. As shown in Fig. 12 of Appendix. B.3, injecting the token-level weight variance noise does not trigger an abnormal increase in the training-inference discrepancy. The modification of Eq. (21) on the original optimization objective can be expressed as,

$$\Delta \mathcal{J}_{\text{var}}(\theta) = \mathcal{J}_{\text{var}}(\theta) - \mathcal{J}_{\text{Seq}}$$

$$= \mathbb{E}_{\text{train}} \left[ \sum_{i,t} Y_{i,t}(\theta) \left( \xi_{i,t} - 1 \right) \right] \tag{22}$$

$$\simeq 0$$

The last approximation is due to the independence of $\xi_{i,t} - 1$ and $Y_{i,t}(\theta)$ (see Appendix A.2). These additional experiments demonstrate that the *biased* distortion is the primary driver of discrepancy growth, as the biased objective-level spurious signals can be persistently exploited.

### 5.2. Monitoring objective-level hacking

In Fig. 6, we show the growth of objective-level signals in Eq. (20) as direct numerical evidence for the occurrence of hacking. We define $J$ as

$$J \equiv \sum_{i,t} \hat{A}_{i,t} \times (\varphi_{i,t} - 1) \tag{23}$$

as an approximate to Eq. (20). We select the sequence-level clipping experiment and the experiment with $\delta = 2$ distortion in Fig. 5 as illustrative examples. Accordingly, we fixed $\delta = 2$ in $\varphi_{i,t}$ of Eq. (23). Compared to the original setup (Fig. 6a), the weight distortion triggers an abnormal growth in such objective-level signal (Fig. 6b). This provides evidence that spurious objective-level signals intervene in the optimization process.

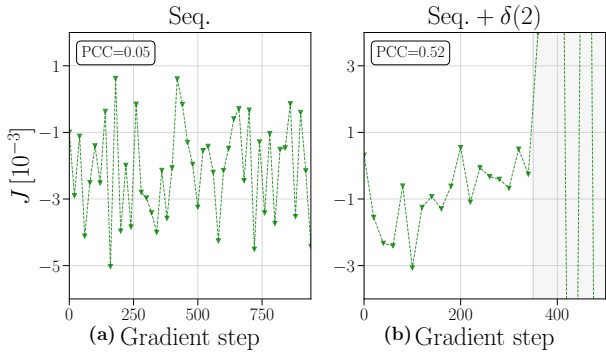

*Figure 6.* **Evolution of the additional optimization objective surrogate during training.** Green dots represent $J$ (defined in Eq. (23)) at each step. The PCC between $J$ and the gradient step is computed and shown in the top-left corner. In panel (b), the PCC is computed using only the data points outside the gray-shaded region.

### 5.3. Positive feedback loop of instability

When training instability occurs, we observe that the resulting degradation in model capabilities is often irreversible. Switching to earlier checkpoints or altering training data batches fails to prevent the onset of collapse (see also Ref. (Zheng et al., 2025b)). This irreversibility can be naturally explained by the *positive feedback loop between training-inference discrepancy and objective-level hacking*.

In Fig. 7, we characterize the discrepancy for tokens across different probability ranges. Low-probability tokens exhibit a wider dispersion from $\rho_{i,t} = 1$, indicating a more significant training-inference mismatch. Furthermore, the importance weight $\rho_{i,t}$ of low-probability tokens consistently decreases as training progresses, which arises from statistical effects induced by survival bias (see Appendix C.2). The divergence of $\rho_{i,t}$ across different probability ranges further exacerbates objective-level hacking, which in turn drives the continued growth of the discrepancy. This posi-

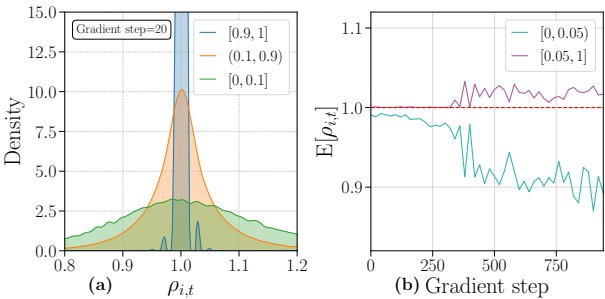

*Figure 7.* **Token-level characterization of training-inference discrepancy.** Results are from token-level clipping experiment in Fig. 2. (a) shows the distribution of $\rho_{i,t}$ for tokens within different probability ranges; (b) illustrates how the mean of $\rho_{i,t}$ within different probability ranges evolves over the training process. Legends shows the interval of $\pi_{\text{train}}$. The red dashed line marks $\rho_{i,t} = 1$.

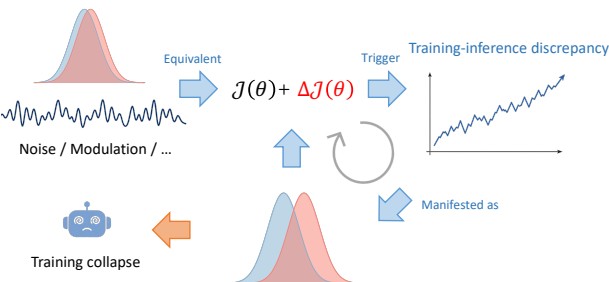

*Figure 8.* **Illustration of positive feedback loop of training collapse.** Noise or token-level modulation introduces spurious objective-level signals, increasing training-inference discrepancy and forming a positive feedback loop with objective-level hacking that ultimately leads to training collapse.

tive feedback loop further leads to the irreversibility of the model's training collapse (Fig. 8).

### 5.4. Contrasting training-inference mismatch dynamics in dense and MoE models

Compared with dense models, MoE architectures have been reported to face more severe training instability during RLVR training (Zheng et al., 2025b; Ma et al., 2025). In this section, we provide an explanation for this phenomenon through the objective-level hacking lens: the training-inference mismatch exhibits a markedly faster growth in MoE models than in dense models during training, thereby amplifying training instability in MoE models.

The theoretical formulation of the objective-level hacking in Section 3 does not rely on MoE-specific assumptions, making it applicable to characterizing the dynamics of dense models as well. We first extend the empirical analysis to dense models, using DeepSeek-R1-Distill-Qwen-7B (Guo et al., 2025) as base model and a different training set, Skywork-OR1-RL-Data (He et al., 2025), to further exam-

ine the generality of the objective-level hacking framework under broader training scenarios. Following the experimental design adopted for the MoE models, we construct four comparative training settings: a vanilla setup with token-level clipping and its TIS-corrected variant to examine hacking induced by the initial training-inference discrepancy; a sequence-level clipping setup serving as a comparison for assessing the hacking from token-level clipping; and a sequence-level clipping setup with injected token-level weight distortion ($\delta = 3$) to test hacking from injected token-level weight distortion. Detailed training configurations are provided in Appendix E.

The training-inference mismatch dynamics of dense models are shown in Fig. 18. As observed in the MoE setting, dense models also exhibit abnormal growth in training-inference mismatch, and both TIS and sequence-level clipping help mitigate this growth. Injecting token-level distortion likewise induces mismatch growth, accompanied by degradation of model performance.

It is worth noting that mismatch growth in dense models is far less severe than in MoE models. Dense models show only a slight mismatch increase (from $1.059 \times 10^{-2}$ to $1.081 \times 10^{-2}$; see Fig. 18), while MoE models exhibit a much larger increase over the same range of gradient steps (from $1.418 \times 10^{-2}$ to $2.664 \times 10^{-2}$; see Fig. 2). This offers a complementary explanation for the severe training instability of MoE models: *in addition to their larger initial training-inference mismatch, MoE models exhibit substantially faster mismatch growth during training than dense models.* The faster mismatch growth may stem from expert activation inconsistency in MoE models, where different experts can be activated during training and inference, thereby creating more room for training-inference mismatch to grow (Dai et al., 2022; Ma et al., 2025; Zhang et al., 2025).

## 6. Related works

In response to model collapse, mitigation strategies have been explored along several dimensions, most at the data, algorithmic, and infrastructure levels.

**Filtering low-quality data.** Because training samples are generated by the model itself, elaborate supervision of sample quality is limited. A prevailing view holds that low-quality trajectories can destabilize optimization. To mitigate this, sequence-level filtering is applied, such as removing highly repetitive samples (Chen et al., 2025) and discarding "void" turns in tool-augmented reasoning (Xue et al., 2026).

**Mitigating model collapse from algorithmic perspectives.** In RLVR, mismatch between rollout and training policies is widely regarded as a key source of instability. Several algorithms have been proposed to address this issue. TIS in-

corporates the mismatch into the optimization objective via truncated importance sampling (Yao et al., 2025), whereas IcePop further introduces clip mechanism to the corresponding importance weights (Zhao et al., 2025). Sequence-level masked importance sampling has been proposed to further mitigate instability (Liu et al., 2025a). Beyond importance sampling, another line of work aims to improve stability through the design of loss aggregation. GSPO employs sequence-level aggregation with sequence-level clipping to enhance robustness (Zheng et al., 2025b). GMPO adopts geometric-mean aggregation to attenuate extreme updates and reduce pathological gradients (Zhao et al., 2026).

**Improving rollout design to reduce mismatch.** Another line of work seeks to reduce rollout–training mismatch by refining the rollout engine itself. Examples include increasing numerical precision in selected components to improve computational accuracy (Chen et al., 2025), and introducing batch-invariant attention to realize a truly on-policy RLVR setup (He & Lab, 2025). In addition, during MoE training, rollout routing replay is employed to align the activated experts (Ma et al., 2025). It should be noted that such modifications typically entail additional computational and systems overhead.

## 7. Summary

In this work, we investigate a key training dynamic behind the instability for MoE model training: the mechanism of abnormal growth in training-inference discrepancy. Through theoretical and empirical studies, we attribute this to the effect of *objective-level hacking* caused by token-level credit misalignment. We provide a principled formulation that unifies the various causes of training instability into a unified framework.

While *reward-level* spurious signals have been widely discussed in RLVR, our study proposes that *objective-level* spurious signals can also pose a threat to training stability. This finding provides a new perspective on the stability analysis of RLVR algorithms, suggesting that we need to carefully consider the potential side effects of token-level modulations.

It is worth noting that RLVR training is a complex multi-system interaction process. Factors such as the choice of model, training data, rollout engine, and even hardware configurations may each induce different forms of training instability (Zhao et al., 2025; Liu et al., 2025a). Therefore, our theoretical explanation does not represent the full picture of model collapse; diverse mechanisms and processes may also contribute to this phenomenon. We hope that our analysis and theoretical perspective can provide useful insights toward improving the interpretability and controllability of model collapse in RLVR.

## Impact Statement

Our work focuses on training instability in reinforcement learning with verifiable rewards for large language models with Mixture-of-Experts architectures. We provide a systematic analysis of the training dynamics and identify several previously underexplored characteristics of such instabilities, including a correlation between abnormal growth in training-inference discrepancy and token credit misalignment. By improving the understanding of training dynamics, our work helps inform the design of more robust reinforcement learning algorithms for Mixture-of-Experts models. There are many potential societal consequences of our work, none which we feel must be specifically highlighted here.

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

# A. Proofs and Derivations

## A.1. Derivations of implicit bias in optimization objective

We provide a detailed derivations of Eq.(10) to illustrate how token-level weight distortion introduces bias into the optimization objective. We begin the derivation from the optimization objective of Group Relative Policy Optimization (GRPO) (Shao et al., 2024), which reads,

$$
\mathcal{J}_{\mathrm{GRPO}}(\theta) = \mathbb{E}_{q \sim \mathcal{D},\, \{o_i\}_{i=1}^{G} \sim \pi_{\mathrm{train}}(\cdot|q;\theta_{\mathrm{old}})}
$$
$$
\left[ \frac{1}{G} \sum_{i=1}^{G} \frac{1}{|o_i|} \sum_{t=1}^{|o_i|} \left( \min\left( r_{i,t}(\theta)\, \hat{A}_{i,t},\ \mathrm{clip}(r_{i,t}(\theta),\, 1-\varepsilon,\, 1+\varepsilon)\, \hat{A}_{i,t} \right) \right) \right] . \tag{24}
$$

Ideally, we expect the sampled responses to follow $\{o_i\}_{i=1}^{G} \sim \pi_{\mathrm{train}}(\cdot|q;\theta_{\mathrm{old}})$ like in Eq. (24). However, in practice, discrepancies may arise between the model used for rollout generation and the one used during training. Therefore, we consider the following three policies: (1) $\pi_{\mathrm{train}}(\cdot|\theta_{\mathrm{old}})$ (the "old" policy model during model training); (2) $\pi_{\mathrm{train}}(\cdot|\theta)$ (the "current" policy model during model training); (3) $\pi_{\mathrm{infer}}(\cdot|\theta_{\mathrm{old}})$ (the model used to collect responses in rollout generation). Define the per-token ratios,

$$
r_{i,t} \equiv \frac{\pi_{\mathrm{train}}(o_{i,t} \mid q,\, o_{i,<t};\theta)}{\pi_{\mathrm{train}}(o_{i,t} \mid q,\, o_{i,<t};\theta_{\mathrm{old}})}, \qquad \rho_{i,t} \equiv \frac{\pi_{\mathrm{train}}(o_{i,t} \mid q,\, o_{i,<t};\theta_{\mathrm{old}})}{\pi_{\mathrm{infer}}(o_{i,t} \mid q,\, o_{i,<t};\theta_{\mathrm{old}})} . \tag{25}
$$

Here, we first slightly reorganize the notation for clarity. For brevity, set

$$
X_{i,t}(\theta) \equiv \frac{r_{i,t}(\theta)\, \hat{A}_{i,t}}{G \cdot |o_i|} . \tag{26}
$$

We write $\mathbb{E}_{\mathrm{train}}[\cdot]$ for expectation over $q \sim \mathcal{D}$, $\{o_i\} \sim \pi_{\mathrm{train}}(\cdot \mid q;\theta_{\mathrm{old}})$, and $\mathbb{E}_{\mathrm{infer}}[\cdot]$ for expectation over $q \sim \mathcal{D}$, $\{o_i\} \sim \pi_{\mathrm{infer}}(\cdot \mid q;\theta_{\mathrm{old}})$. Assuming an ideal setting where all samples are drawn from $\pi_{\mathrm{train}}(\cdot|\theta_{\mathrm{old}})$ without any practical bias, the GRPO optimization objective takes the form,

$$
\mathcal{J}(\theta) = \mathbb{E}_{q \sim \mathcal{D},\, \{o_i\} \sim \pi_{\mathrm{train}}(\cdot|\theta_{\mathrm{old}})} \left[ \frac{1}{G} \sum_{i=1}^{G} \frac{1}{|o_i|} \sum_{t=1}^{|o_i|} r_{i,t}(\theta)\, \hat{A}_{i,t} \right] = \mathbb{E}_{\mathrm{train}} \left[ \sum_{i,t} X_{i,t}(\theta) \right] . \tag{27}
$$

In practice, the responses are actually sampled from $\pi_{\mathrm{infer}}(\cdot|\theta_{\mathrm{old}})$. Consequently, the effective optimization objective in practice should be rewritten as[3]:

$$
\mathcal{J}'(\theta) = \mathbb{E}_{\mathrm{infer}} \left[ \sum_{i,t} X_{i,t}(\theta) \right] \simeq \mathbb{E}_{\mathrm{train}} \left[ \sum_{i,t} X_{i,t}(\theta)\, \rho_{i,t}^{-1} \right] , \tag{28}
$$

and their difference is

$$
\Delta \mathcal{J}(\theta) \equiv \mathcal{J}'(\theta) - \mathcal{J}(\theta) \simeq \mathbb{E}_{\mathrm{train}} \left[ \sum_{i,t} X_{i,t}(\theta)\, (\rho_{i,t}^{-1} - 1) \right] . \tag{29}
$$

Note that for any $(q,\, o_{i,<t})$, we have,

$$
\mathbb{E}_{o_{i,t} \sim \pi_{\mathrm{train}}(\cdot|q,\, o_{i,<t};\theta_{\mathrm{old}})} \left[ \rho_{i,t}^{-1} \right] = \mathbb{E}_{o_{i,t} \sim \pi_{\mathrm{train}}(\cdot|q,\, o_{i,<t};\theta_{\mathrm{old}})} \left[ \frac{\pi_{\mathrm{infer}}(o_{i,t} \mid q,\, o_{i,<t};\theta_{\mathrm{old}})}{\pi_{\mathrm{train}}(o_{i,t} \mid q,\, o_{i,<t};\theta_{\mathrm{old}})} \right] = 1 . \tag{30}
$$

---

[3] The approximation symbol $\simeq$ in Eq. (28) originates from the violation of the independence assumption underlying importance sampling. Strictly speaking, when applying importance sampling $\mathbb{E}_{x \sim q}[f(x)] = \mathbb{E}_{x \sim p}\left[ f(x) \frac{q(x)}{p(x)} \right]$, it is required that the samples $\{f(x_i)\}_p$ and $\{f(x_i)\}_q$ are independently determined across samples. However, in GRPO and related algorithms that rely on advantage-based objectives, the computation of advantages introduces batch-level dependencies among samples. The approximation symbol used here is equivalent to adopting a *"surrogate" objective* under a first-order approximation (Schulman et al., 2017).

Hence $\mathbb{E}_{\text{train}}[\rho_{i,t}^{-1} - 1] = 0$. Starting from (29), we get

$$\Delta \mathcal{J}(\theta) \simeq \sum_{i,t} \mathbb{E}_{\text{train}}\big[X_{i,t}(\theta)\,(\rho_{i,t}^{-1} - 1)\big] = \sum_{i,t} \text{Cov}_{\text{train}}\big(X_{i,t}(\theta),\ \rho_{i,t}^{-1}\big). \tag{31}$$

where Cov denotes covariance operator,

$$\text{Cov}_{\text{train}}(U, V) \equiv \mathbb{E}_{\text{train}}\big[(U - \mathbb{E}_{\text{train}}[U])\,(V - \mathbb{E}_{\text{train}}[V])\big] = \mathbb{E}_{\text{train}}[UV] - \mathbb{E}_{\text{train}}[U]\,\mathbb{E}_{\text{train}}[V]. \tag{32}$$

Consequently, our practical optimization objective can be *interpreted* as,

$$\mathcal{J}'(\theta) = \mathcal{J}(\theta) + \Delta\mathcal{J}(\theta). \tag{33}$$

The additional term $\Delta\mathcal{J}(\theta)$ effectively perturbs the original optimization objective, acting as an objective-level spurious signal that interferes with the model's behavior. For instance, in pursuit of increasing $\Delta\mathcal{J}(\theta)$, the model may be inclined to optimize in the following ways: when $\hat{A}_{i,t} > 0$ the model is incentivized to generate more tokens with larger $\rho_{i,t}^{-1} = \pi_{\text{infer}}/\pi_{\text{train}}$; when $\hat{A}_{i,t} < 0$, the opposite holds. This may introduce pathological tendencies into the model's inference process, leading to a growth in distribution mismatch.

If we incorporate a token-level reweighting factor $\phi_{i,t}$, the effective optimization objective becomes,

$$\mathcal{J}_\phi(\theta) = \mathbb{E}_{\text{train}}\left[ \sum_{i,t} X_{i,t}(\theta)\,\phi_{i,t} \right] \equiv \mathcal{J}(\theta) + \Delta_\phi \mathcal{J}(\theta). \tag{34}$$

where,

$$\Delta_\phi \mathcal{J}(\theta) = \mathcal{J}_\phi(\theta) - \mathcal{J}(\theta) = \mathbb{E}_{\text{train}}\left[ \sum_{i,t} X_{i,t}(\theta)\,(\phi_{i,t} - 1) \right]. \tag{35}$$

This implies that, on top of the vanilla objective, the model may be incentivized to increase $\Delta_\phi \mathcal{J}(\theta)$. If $\phi_{i,t}$ is strongly correlated with token probabilities, like,

$$\phi_{i,t} = \begin{cases} 1 + \epsilon, & \text{if } o_{i,t} \in S, \\ 1, & \text{otherwise}. \end{cases} \tag{36}$$

where $\epsilon > 0$ and $S$ denotes a set whose membership is strongly correlated with token probabilities. To increase $\Delta_\phi \mathcal{J}(\theta)$, the model is incentivized to favor tokens in $S$ when $\hat{A}_{i,t} > 0$ and to suppress them when $\hat{A}_{i,t} < 0$. This unintended pursuit of the correlation between $\phi_{i,t}$ and $\phi_{i,t} - 1$ leads to pathological behavior in the model's inference process, ultimately affecting the stability of training. We compute the dynamics of such objective-level signals (Fig 6), serving as empirical evidence supporting the above analysis.

### A.2. Proofs of Eq. (22)

We aim to prove,

$$\mathbb{E}_{\text{train}}\left[ \sum_{i,t} Y_{i,t}(\theta)\,(\xi_{i,t} - 1) \right] \simeq 0. \tag{37}$$

The complete form of the left-hand side can be written as,

$$\mathbb{E}_{q \sim \mathcal{D}, \{o_i\} \sim \pi_{\text{train}}(\cdot|\theta_{\text{old}})}\left[ \frac{1}{G} \sum_{i=1}^{G} \frac{1}{|o_i|} \sum_{t=1}^{|o_i|} \min\left( s_{i,t}(\theta)\,\hat{A}_{i,t}, \text{clip}\left(s_{i,t}(\theta), 1 - \epsilon, 1 + \epsilon\right) \hat{A}_{i,t} \right) \times (\xi_{i,t} - 1) \right], \tag{38}$$

where $\xi_{i,t}$ is independently sampled from a Gaussian distribution $\mathcal{N}(1, \sigma^2)$.

For the term

$$\frac{1}{G} \sum_{i=1}^{G} \frac{1}{|o_i|} \sum_{t=1}^{|o_i|} \min \left( s_{i,t}(\theta) \hat{A}_{i,t}, \mathrm{clip}\left( s_{i,t}(\theta), 1 - \epsilon, 1 + \epsilon \right) \hat{A}_{i,t} \right) \times (\xi_{i,t} - 1) \,, \tag{39}$$

the operator $\frac{1}{G} \sum_{i=1}^{G} \frac{1}{|o_i|} \sum_{t=1}^{|o_i|}$ represents a Monte Carlo approximation to the expectation operator over a set of tokens generated by models $\{o_{i,t}\}$. We denote it as $\mathbb{E}_{o_{i,t}}$. Thus, Eq. (39) can be approximated as,

$$\mathbb{E}_{o_{i,t}} \left[ \min \left( s_{i,t}(\theta) \hat{A}_{i,t}, \mathrm{clip}\left( s_{i,t}(\theta), 1 - \epsilon, 1 + \epsilon \right) \hat{A}_{i,t} \right) \times (\xi_{i,t} - 1) \right] \,. \tag{40}$$

Note that the variable $\xi_{i,t}$ follows a normal distribution $\mathcal{N}(1, \sigma^2)$, so $(\xi_{i,t} - 1)$ follows a normal distribution $\mathcal{N}(0, \sigma^2)$. Therefore, the variable $(\xi_{i,t} - 1)$ is *independent* of the variable $\min \left( s_{i,t}(\theta)\hat{A}_{i,t}, \mathrm{clip}\left( s_{i,t}(\theta), 1 - \epsilon, 1 + \epsilon \right) \hat{A}_{i,t} \right)$. Thus, we have

$$\begin{aligned}
&\mathbb{E}_{o_{i,t}} \left[ \min \left( s_{i,t}(\theta) \hat{A}_{i,t}, \mathrm{clip}\left( s_{i,t}(\theta), 1 - \epsilon, 1 + \epsilon \right) \hat{A}_{i,t} \right) \times (\xi_{i,t} - 1) \right] \\
&= \mathbb{E}_{o_{i,t}} \left[ \min \left( s_{i,t}(\theta) \hat{A}_{i,t}, \mathrm{clip}\left( s_{i,t}(\theta), 1 - \epsilon, 1 + \epsilon \right) \hat{A}_{i,t} \right) \right] \times \mathbb{E}_{o_{i,t}} \left[ \xi_{i,t} - 1 \right] \\
&= 0 \,.
\end{aligned} \tag{41}$$

The final equality follows from $\mathbb{E}_{o_{i,t}} \left[ \xi_{i,t} - 1 \right] = 0$. Therefore, we have,

$$\frac{1}{G} \sum_{i=1}^{G} \frac{1}{|o_i|} \sum_{t=1}^{|o_i|} \min \left( s_{i,t}(\theta) \hat{A}_{i,t}, \mathrm{clip}\left( s_{i,t}(\theta), 1 - \epsilon, 1 + \epsilon \right) \hat{A}_{i,t} \right) \times (\xi_{i,t} - 1) \simeq 0 \,, \tag{42}$$

and further

$$\mathbb{E}_{\text{train}} \left[ \sum_{i,t} Y_{i,t}(\theta) (\xi_{i,t} - 1) \right] \simeq 0 \,. \tag{43}$$

## B. Supplementary experiments

### B.1. Supplementary metrics to initial training-inference discrepancy

As an extension of Fig. 2 and Fig. 3, we examine the evolution of key indicators throughout training. In Fig. 9a, we show how the Pearson correlation coefficient (PCC) between $\pi_{\text{train}}$ and $\pi_{\text{infer}}$ evolves throughout the training. In Fig. 9b, we show the evolution of the maximum value of $\rho_{i,t}$ at each training step, which exhibits a similar increasing trend. All these phenomena collectively indicate that the model's original inference mode has been disrupted.

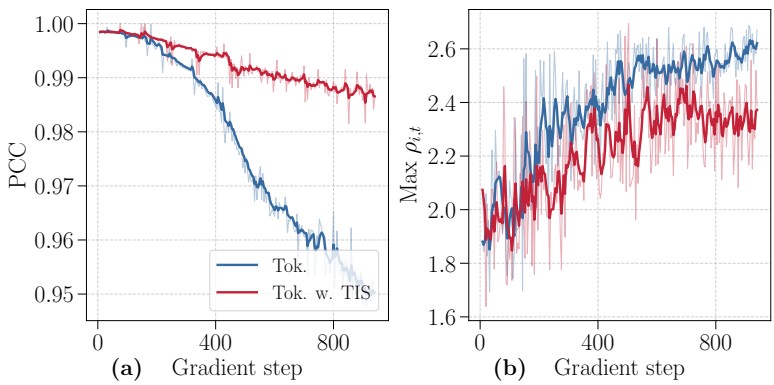

*Figure 9.* **Additional metrics illustrating training-inference discrepancy.** (a) shows the per-step PCC corresponding to $\pi_{\text{train}}(\cdot \mid \theta_{\text{old}})$ and $\pi_{\text{infer}}(\cdot \mid \theta_{\text{old}})$; (b) illustrates the evolution of the maximum $\rho_{i,t}$ at each step.

**B.2. Supplementary experiments to injected token-level weight distortion**

Figure 10 shows the training dynamics corresponding to decreasing the low-probability token weight, where $\delta = 0$ in Eq. (18). As shown in Fig. 10, decreasing the low-probability token weight also induces abnormal growth in the training-inference discrepancy. This demonstrates that *weight distortion*, rather than increasing that weight, is the root cause of the growth.

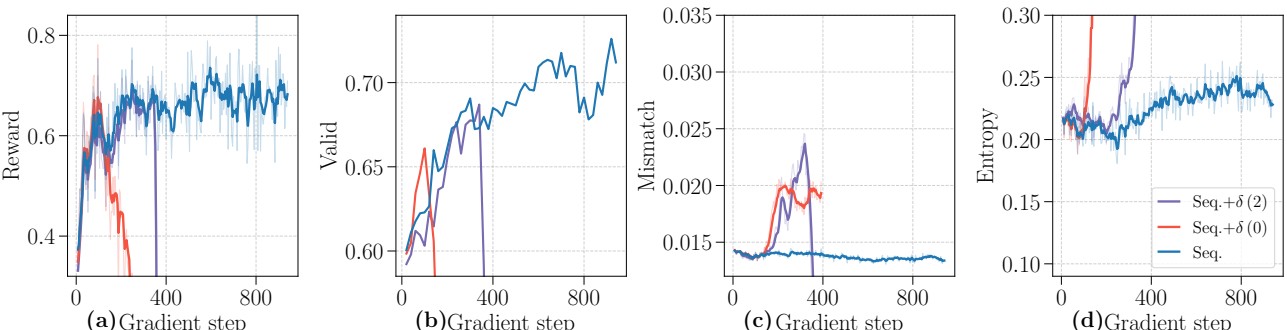

*Figure 10.* **Influence of injected token-level weight distortion (decreasing the token weights).** Blue curves correspond to training dynamics with sequence-level clipping; Purple curves denote the same configuration with injected token-level weight distortion as $\delta = 2$; Red curves denote that with $\delta = 0$.

**B.3. Supplementary experiments to injected token-level weight variance**

Figure 11 illustrates our approach to investigating whether *biased* objective-level signals is the primary driver of the abnormal dynamics underlying training-inference discrepancy. Unlike the biased optimization signal injection in Section 4.4 (see Eq. (20)), we design a variance-based token weight distortion with an objective-level *unbiased* noise injection (see Eq. (22)). This form of weight distortion does not introduce implicit preference toward specific tokens, but instead only increases the dispersion of token weights. This ablation setting enables a clearer understanding of the mechanism and contribution of objective-level hacking to the observed abnormal dynamics.

Figure 12 shows the training dynamics corresponding to injecting variance-based token weight perturbation. The optimization objective is given by Eq. (21), where we set $\xi_{i,t}$ to follow the Gaussian distribution $\mathcal{N}(1, \sigma^2)$ with $\sigma = 0.2$. As shown in Fig. 12, variance-based token weight perturbation does not induce abnormal growth in the training-inference discrepancy.

## C. Token-level characterization of training-inference discrepancy for MoE models

We conduct a detailed investigation of the numerical behavior of the training-inference discrepancy at the token level and analyze its impact on RLVR training. Among all the phenomena, two types of specific numerical phenomena catch our attention. The first phenomenon refers to the *abnormal clustering* of token probabilities in the $\pi_{infer}$ dimension, as shown in Fig. 3, exhibiting a distinct band-like distribution. The second phenomenon is shown in Fig. 7b, where the importance weight of low-probability tokens continuously decreases during training. Broadly speaking, these noises originate from implementation discrepancies between rollout generation and training in RL framework (Yao et al., 2025; Liu et al., 2025a). We further investigate several finer-grained numerical factors associated with these phenomena and briefly discuss their effects on model training.

**C.1. Abnormal clustering of $\pi_{infer}$**

After extensive analysis, we conclude that this abnormal clustering originates from the limited numerical precision of inference engines. In RL framework, efficient rollout engines are employed to generate experiences (Kwon et al., 2023; Zheng et al., 2024). To improve computational efficiency, numerical operations are often performed in the BF16, FP8, or INT8 format (Sheng et al., 2025; Wang et al., 2025; Liu et al., 2025b).

To reproduce this phenomenon, we simulate the process of obtaining token probabilities by applying the softmax function to logits under BF16 precision. In each simulation, we calculate $5 \times 10^4$ token probabilities. We assume that all logits follow a Gaussian distribution, $z \sim \mathcal{N}(0, 1.5^2)$. The size of the virtual vocabulary is set to 2048.

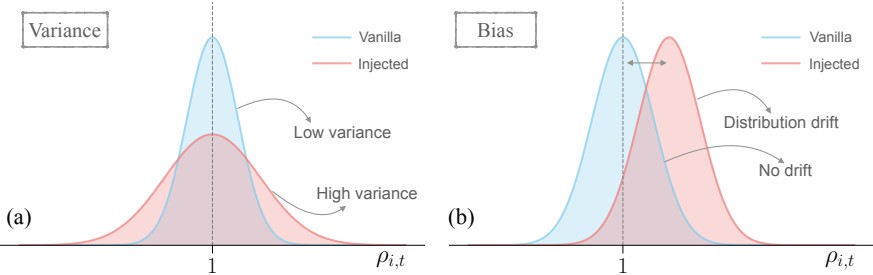

*Figure 11.* **Illustration of variance-based and bias-based token weight distortion injections in Section 5.1.** Figures (a) and (b) provide schematic illustrations of variance-based and bias-based token weight distortion injections, respectively. The distributions represent $\rho_{i,t}$ across output tokens. Ideally, $\rho_{i,t} = 1$, indicating no weight distortion. In practice, numerical effects introduce slight dispersion (denoted as vanilla). In variance-based injection experiments, noise is added to increase the dispersion without introducing systematic bias, while bias-based injection induces a global drift, reflecting additional preference toward specific tokens.

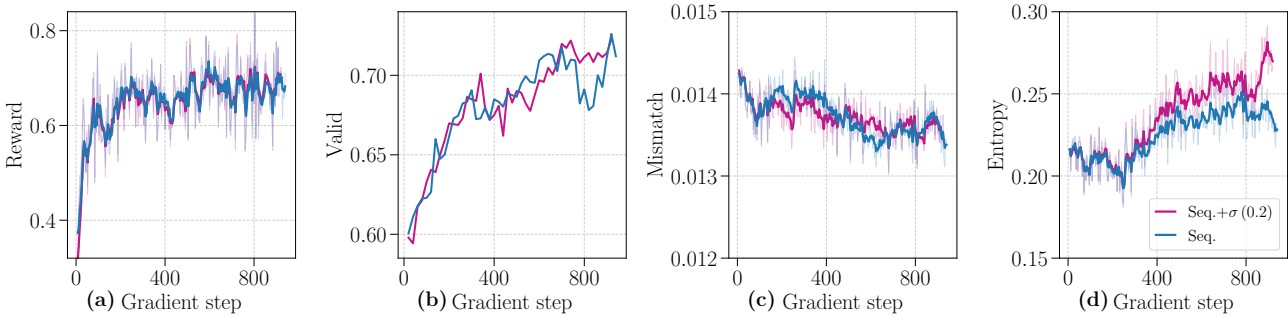

*Figure 12.* **Influence of injected token-level weight variance.** Blue curves correspond to training dynamics with sequence-level clipping. Purple curves denote the same configuration with injected token-level weight variance.

In simulations, we always generate logits in FP32 precision. Then, we compute the softmax under both BF16 and FP32 precision to get the token probability distribution. Based on the BF16 probability distribution, we perform sampling to obtain token indices, which allows us to extract two corresponding probabilities for each selected token,

- Token probability (BF16): the probability computed in BF16 precision,

- Token probability (FP32): the probability at the same index computed in FP32 precision.

This procedure is repeated under multiple random seeds, and the distributions of the two sets of probabilities are summarized and presented in Fig. 13. Compared to FP32, token distribution computed in BF16 exhibit localized discretization: subsets of tokens concentrate at specific probability levels, producing sharp spikes in the distribution. Varying the random seed leaves the locations and amplitudes of these discretization peaks nearly invariant, indicating that they are not attributable to stochastic numerical error or sampling noise, but rather to systematic artifacts of finite-precision BF16 computation.

Such discretization artifacts introduce numerical noise and induce offline effects in reinforcement learning. Detailed statistics are reported in Tab. 2. Concretely, the softmax function is defined as,

$$p_i = \frac{\exp(z_i)}{\sum_{j=1}^{V} \exp(z_j)}, \tag{44}$$

where $z_i$ denotes the logits. When computed in BF16, the probability is quantized as

$$\tilde{p}_i = Q_{\text{bf16}}\left(\frac{e^{Q_{\text{bf16}}(z_i)}}{\sum_{j=1}^{V} e^{Q_{\text{bf16}}(z_j)}}\right), \tag{45}$$

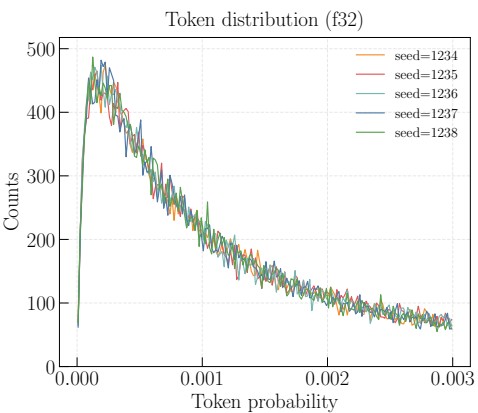 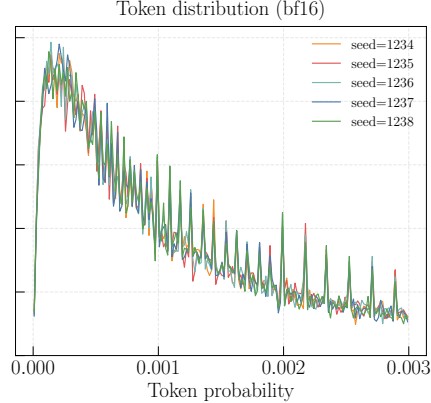

*Figure 13.* **Discretization artifacts of distribution mismatch.** The left panel shows the token probability distribution of sampled tokens computed in float32 precision. The right panel presents the corresponding token probability distribution computed in bfloat16 precision for the same sampled tokens. Different colors indicate results obtained under different random seeds.

where $Q_{\mathrm{bf16}}(\cdot)$ denotes the BF16 quantization operator. Since BF16 effectively provides only limited decimal digits of precision, probabilities in the low-probability regime are constrained to sparse discrete levels, preventing a smooth continuous representation. The softmax function fundamentally applies an exponential mapping followed by normalization, which involves summing a large number of exponentials and thereby amplifies this discreteness. As a result, many probabilities collapse onto a limited set of discrete values, producing non-smooth artifacts in the probability distribution.

*Table 2.* Ratio of token probability between float32 and bf16 and KL divergence statistics

| Seed | Mean | Std | Max | Min | KL Divergence |
|---|---|---|---|---|---|
| 1234 | 1.0000 | 0.0048 | 1.0190 | 0.9808 | 0.046838 |
| 1235 | 1.0000 | 0.0048 | 1.0193 | 0.9814 | 0.019425 |
| 1236 | 1.0000 | 0.0048 | 1.0188 | 0.9818 | 0.015542 |
| 1237 | 1.0000 | 0.0048 | 1.0189 | 0.9815 | 0.014063 |
| 1238 | 1.0000 | 0.0048 | 1.0185 | 0.9818 | 0.025232 |
| Average | 1.0000 | 0.0048 | 1.0189 | 0.9815 | 0.024220 |

This numerical phenomenon implies that, as training progresses, the training-inference discrepancy may increase alongside this phenomenon, ultimately disrupting the model's inference pattern. This observation also offers insights into maintaining stable model training under low numerical precision.

### C.2. Drift of importance weight for low-probability tokens

As shown in Fig 7b, we find that the $\rho_{i,t}$ values of low-probability tokens deviate from 1, exhibiting a consistent downward bias. We point out that such drift actually arises from a form of *survival bias*. Intuitively, during the rollout generation process, tokens are sampled according to $\pi_{\mathrm{infer}}$. Therefore, *the sampled tokens are unlikely to come from tokens with extremely low probabilities under $\pi_{\mathrm{infer}}$; otherwise, they would not have been sampled*. In expectation, the average $\pi_{\mathrm{infer}}$ of this sampled batch tends to be higher than that of another unrelated distribution $\pi_{\mathrm{train}}$. This effect is more pronounced in the low-probability region, indicating that survival bias essentially introduces an inherent form of distribution drift.

The existence of this phenomenon creates a *positive feedback loop* between weight distortion and discrepancy (see Fig. 8). As the discrepancy intensifies, survival bias becomes more pronounced, leading to more severe weight distortion for low-probability tokens. This increased distortion, in turn, drives further growth of the discrepancy. This mechanism often makes the occurrence of training instability *irreversible*.

Considering that the distributional discrepancy between $\pi_{\mathrm{train}}$ and $\pi_{\mathrm{infer}}$ is difficult to quantify precisely, we provide the following alternative argument to partially account for the observed bias. We have,

$$\mathbb{E}_{o_{i,t}\sim\pi_{\mathrm{infer}}(\cdot|q,o_{i,<t};\theta_{\mathrm{old}})}\left[\rho_{i,t}^{-1}\right]\geq 1\,. \tag{46}$$

where,

$$\rho_{i,t}^{-1} = \frac{\pi_{\text{infer}}(o_{i,t} \mid q, o_{i,<t}; \theta_{\text{old}})}{\pi_{\text{train}}(o_{i,t} \mid q, o_{i,<t}; \theta_{\text{old}})} . \tag{47}$$

When samples are drawn from the $\pi_{\text{infer}}$, the resulting values of $\pi_{\text{infer}}$ tend to be larger on average than those from an unrelated distribution $\pi_{\text{train}}$. A brief proof of the above equation is given in Proof C.2.

*Proof.* Consider the functions

$$f(x) = \frac{p(x)}{\sqrt{q(x)}}, \quad g(x) = \sqrt{q(x)}.$$

Then

$$\int f(x)g(x)\,dx = \int p(x)\,dx = 1.$$

By the Cauchy–Schwarz inequality,

$$\left( \int f(x)g(x)\,dx \right)^2 \leq \left( \int f(x)^2 dx \right)\left( \int g(x)^2 dx \right).$$

That is,

$$1^2 \leq \left( \int \frac{p(x)^2}{q(x)}\,dx \right) \cdot \left( \int q(x)\,dx \right).$$

Since $\int q(x)\,dx = 1$, we obtain

$$\int \frac{p(x)^2}{q(x)}\,dx \geq 1.$$

Therefore,

$$\mathbb{E}_{x \sim p}\left[ \frac{p(x)}{q(x)} \right] \geq 1,$$

with equality if and only if $p = q$ everywhere. $\qquad\square$

## D. Training-inference discrepancy in relation to other training dynamics metrics

As a training metric in the RLVR, this section discusses the relationship between training-inference discrepancy and other metrics used to monitor training dynamics, such as token entropy and gradient norm, to gain deeper insights into reinforcement learning training dynamics.

Overall, when training collapse occurs, training-inference discrepancy shows anomalous behavior consistent with other metrics. Notably, it often emerges as the *earliest* indicator of impending collapse. This highlights its value as an *early-warning* signal for monitoring training instability, and further suggests that training–inference discrepancy may play a more fundamental role in collapse dynamics, with anomalies in other metrics arising as correlated effects.

We present the extended training dynamics in Figs. 15, 16, and 17 and detailed discussions of the relationships between training–inference discrepancy and token entropy as well as gradient norm in Sections D.1 and D.2

### D.1. Relation to token entropy

Token entropy is a metric that is widely used to monitor model states in RLVR, as it reflects the diversity of model outputs, with higher values indicating richer token diversity (Yu et al., 2025). Prior studies have shown that during the training token entropy often decreases progressively and may eventually collapse to very low levels, signaling a loss of exploration capability. The phenomenon is commonly referred to as *token-entropy collapse* (Yu et al., 2025; He et al., 2025; Cui et al., 2025). Meanwhile, the growth of the training-inference discrepancy is observed to be correlated with token-entropy collapse, as evidenced by the fact that introducing TIS correction alleviates token-entropy collapse in dense model training (Yao et al., 2025).

In MoE training, we observe a more diverse set of correlation patterns between the two metrics. Overall, as the training-inference mismatch increases, it is often accompanied by anomalies in token entropy. Such anomalies may manifest either

as token-entropy collapse (see Fig. 2) or as a pronounced increase in token entropy (see Fig. 5). Whether manifesting as collapse or surge, such anomalies consistently indicate a substantial abnormality in the model's inference patterns. This relation suggests that an increasing training-inference discrepancy indicates not only a growing mismatch between training and inference token distributions, but also a *pathological deviation* in the model's inference pattern itself.

### D.2. Relation to grad norm

Gradient norm is another important metric for monitoring training stability. As shown in Fig. 15-17, when training-inference discrepancy exhibits anomalous behavior, we typically observe corresponding anomalies in the gradient norm.

Moreover, we find that training-inference discrepancy becomes anomalous at an earlier stage, serving as a more effective early-warning signal for monitoring training health. As shown in Fig. 14, we present the dynamics of the training-inference discrepancy, token entropy, and gradient norm during the early stage of training. It can be observed that *when the training-inference discrepancy increases noticeably to a detectable level, neither the token entropy nor the gradient norm exhibits clear abnormalities*. In contrast, in the later stage of training (Fig. 17), when training collapse occurs, all three dynamics show anomalous behaviors. This observation suggests, to some extent, that the growth of the training-inference discrepancy is a more fundamental signal of training instability, and that the accumulation and amplification of this discrepancy trigger abnormalities in other training metrics.

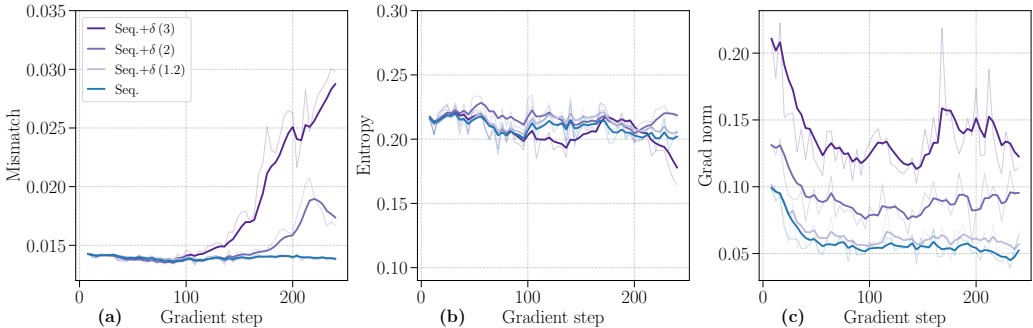

*Figure 14.* **Early stage of training dynamics of Fig. 5 (injected token-level weight distortion).**

### D.3. Extended training dynamics

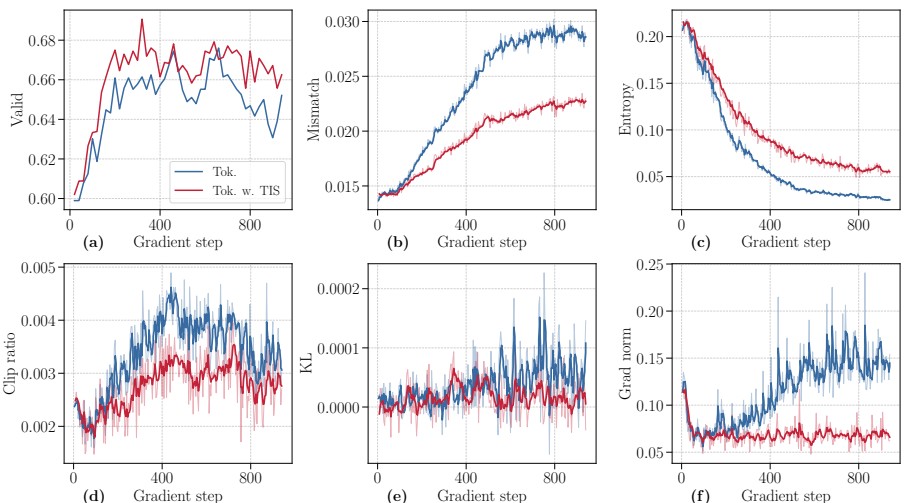

*Figure 15.* **Extended training dynamics of Fig. 2 (initial training-inference discrepancy).** The subfigures respectively show (a) validation accuracy, (b) mismatch (training–inference discrepancy), (c) token entropy, (d) token clipping ratio, (e) KL divergence with the base model, and (f) gradient norm.

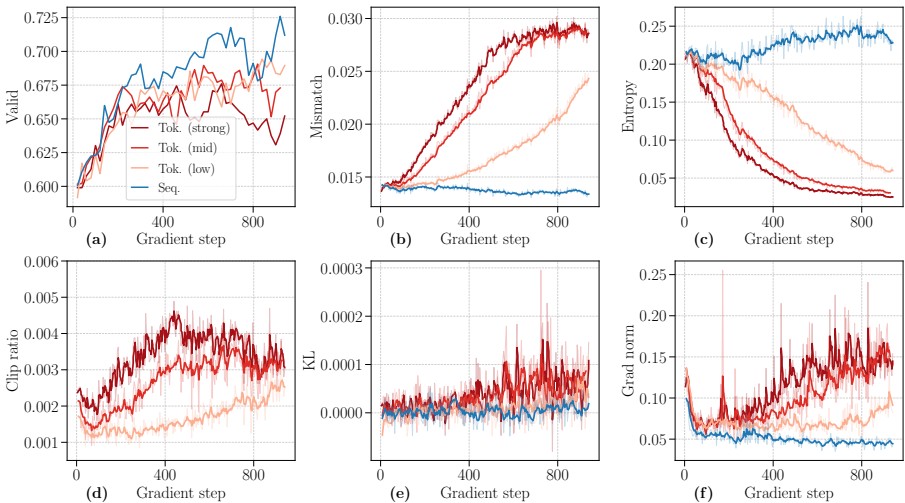

*Figure 16.* **Extended training dynamics of Fig. 4 (token-level clipping).** The metrics shown are the same as in Fig. 15.

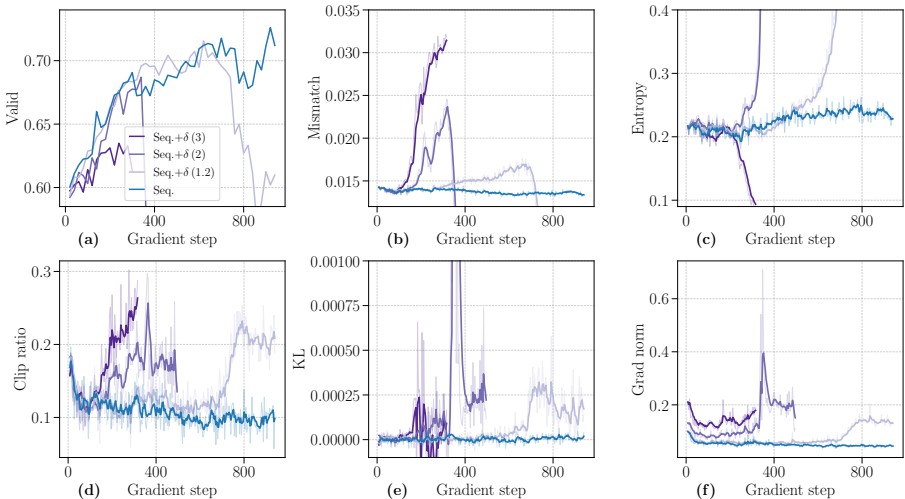

*Figure 17.* **Extended training dynamics of Fig. 5 (injected token-level weight distortion).** The metrics shown are the same as in Fig. 15.

## E. Extension to dense models

We conduct the additional empirical analysis to dense models, based on DeepSeek-R1-Distill-Qwen-7B (Guo et al., 2025), using Skywork-OR1-RL-Data (He et al., 2025) as the training set. Following the experimental design adopted for the MoE models, we consider the following four comparative training settings:

- **(Tok.)**: A vanilla GRPO setup with token-level clipping.

- **(Tok. w. TIS)**: A TIS-corrected variant of the token-level clipping setup, used to examine hacking induced by the initial training-inference discrepancy.

- **(Seq.)**: A sequence-level clipping setup, serving as a comparison for assessing the hacking from token-level clipping.

- **(Seq. + $\delta(3)$)**: A sequence-level clipping setup with injected token-level weight distortion, where $\delta = 3$ (see Eq. 18), used to test hacking from injected token-level weight distortion.

The detailed training configuration is as follows. The experiments are implemented with `verl` framework (Sheng et al., 2025), with vLLM serving as the inference backend (Kwon et al., 2023) and Megatron as the training backend (Shoeybi

et al., 2019). At each training step, we sample 128 problems and generate 16 responses per problem, followed by 4 parameter update steps. The maximum response length for each response is set to 8K. For setup with token-level clipping, we use left and right clipping ranges of 0.2; for setup with sequence-level clipping, we set the left and right clipping ranges to 3e-4 and 4e-4, respectively (Zheng et al., 2025b). All experiments of dense models are conducted on 4 nodes × 8 A100 GPUs. The training dynamics of dense-model experiments are shown in Fig. 18.

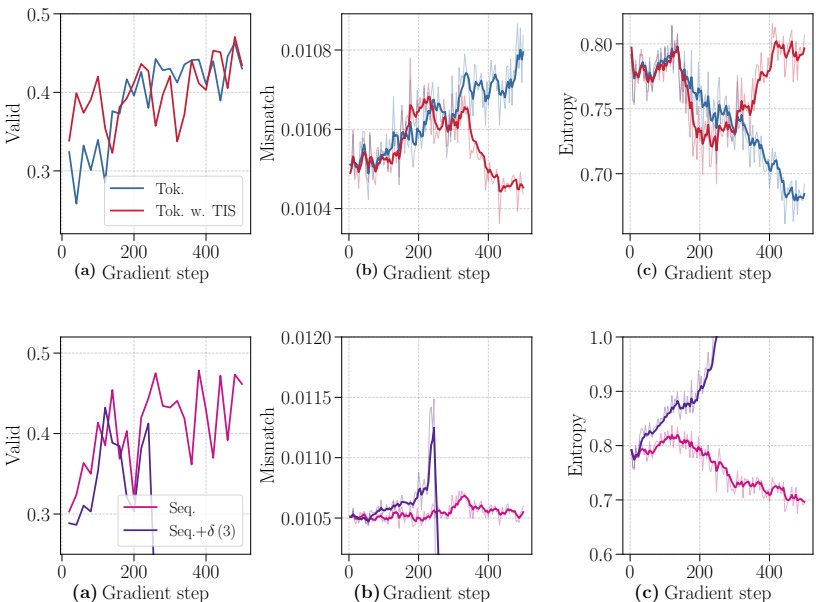

*Figure 18.* **Training dynamics of the dense model DeepSeek-R1-Distill-Qwen-7B.** The training data are from Skywork-OR1-RL-Data (He et al., 2025), and AIME24 is used as the validation set.

## F. Interpretation of stabilization strategies through objective-level hacking lens

We have provided an interpretation of the effectiveness of existing stabilization strategies under the objective-level hacking framework.

- **Data filtering**: Filtering low-quality samples (e.g. repetitive rollouts (Chen et al., 2025); void turns in tool-augmented reasoning (Xue et al., 2026)) is a common stabilization practice. In the objective-level hacking framework, such methods do not introduce sereve MoE instability because they inherently operate at the sequence level, which is safer for MoE models.

- **Algorithms**: Algorithms such as TIS (Yao et al., 2025) or GSPO (Zheng et al., 2025b) do not reduce mismatch at the infrastructure level, but instead mitigate or break the objective-level hacking effect at the objective design level, thereby maintaining training stability.

- **Infrastructure**: Infrastructure-level interventions, such as routing replay (Ma et al., 2025) and batch-invariant attention (He & Lab, 2025) can reduce the initial mismatch, which also mitigates the objective-level hacking effect (see Fig. 8) and helps avoid early entering a positive feedback loop during training.

