# OpenReview forum: "Probing RLVR Training Instability through the Lens of Objective-Level Hacking"
_ICML.cc/2026/Conference — ICML 2026 regular_

### Official Review · Reviewer_yjSN · 2026-03-12

**Soundness:** 2
**Presentation:** 3
**Significance:** 2
**Originality:** 3
**Overall Recommendation:** 3
**Confidence:** 3

**Summary:**

The paper investigates training instability in large language models—specifically Mixture-of-Experts (MoE) architectures—during Reinforcement Learning with Verifiable Rewards (RLVR). The authors propose a novel framework called "Objective-level hacking." They argue that token credit misalignment, caused by training-inference discrepancies and algorithmic choices like token-level clipping, introduces a spurious, biased signal ($\Delta\mathcal{J}(\theta)$) into the optimization objective. Based on empirical analysis using the Qwen3-30B-A3B model, the authors conclude that this objective-level hacking is the root cause of the abnormal growth in training-inference discrepancy and subsequent model collapse.

**Compliance With Llm Reviewing Policy:**

Affirmed.

**Final Justification:**

I thank the authors for their continued effort and additional experiments. However, after careful consideration, I find the core issues remain unresolved.

1. The Universality Gap Remains Unexplained
My central concern was never about terminology ("exponential" vs. "abnormal"), but about a fundamental logical gap. The entire theoretical framework (Eqs. 9-10, 13-15) is derived from generic GRPO formulations with no MoE-specific assumptions — it should apply equally to dense models. Yet the authors' own dense model data shows mismatch barely grows (1.059→1.081), while MoE models exhibit pronounced growth. The authors acknowledge this discrepancy ("mismatch levels differ between dense and MoE") but never explain why.

The most natural explanation is MoE routing inconsistency — recent work (R3) has explicitly shown that ~10% of routers select different experts between training and inference, and 94% of tokens have at least one layer with different expert selection. The paper never engages with this well-documented mechanism, which suggests objective-level hacking may be an indirect mathematical redescription of a routing-driven phenomenon rather than an independent causal mechanism. Notably, the routing-based explanation can account for all key experimental findings — including why TIS helps, why stronger token-level clipping worsens mismatch, why injecting bias into low-probability tokens triggers growth, and why dense models are unaffected — without requiring the objective-level hacking framework.

The authors argue that "focusing on MoE does not imply limited scope," and I agree in principle. However, the issue is not about studying MoE — it is about claiming a universal optimization-level mechanism when the evidence is architecture-specific. If the framework's explanatory power vanishes on dense models, it should be presented as an MoE-specific analysis, not a general theory of RLVR instability.

2. "Irreversible Collapse" — Partially Addressed

I acknowledge that "irreversible collapse" is standard terminology in the community (as used in GSPO), and the extended training data (Table 1, Valid: 65.7→51.6 over 1600 steps) does show sustained degradation, which partially supports the narrative. However, the collapse patterns reported in GSPO and RSPO involve abrupt reward crashes where the model underperforms the base model, whereas the degradation shown here is gradual and the model still performs well above its starting point. The severity gap between the cited phenomena and the paper's own evidence remains notable. I soften this criticism but maintain the observation.

3. Contribution Remains Limited

The authors reframe their contribution as (a) a novel theoretical framework, (b) causal intervention experiments, and (c) practical guidance favoring sequence-level methods. However: (a) the framework's explanatory power is undermined by its failure to distinguish MoE from dense behavior, as discussed above; (b) the injection experiments show that artificial distortion can cause mismatch growth, but do not demonstrate that natural training dynamics do operate through this mechanism rather than through routing inconsistency; (c) the practical recommendation to prefer sequence-level clipping is already the established conclusion of GSPO. Monitoring distribution mismatch is also standard operational practice in modern RLVR pipelines, limiting the novelty of the "early-warning" framing.

I maintain my current score.

**Key Questions For Authors:**

1 Model & Benchmark Stress-Test: Show that this objective-level hacking occurs on at least one other fundamentally different model, and evaluate it on a highly difficult benchmark like HMMT25, AIME25 to prove the collapse is not a byproduct of signal saturation.

2 The "Brakes" Test: Re-introduce the standard KL divergence penalty. Does token-level clipping still cause irreversible discrepancy growth when the policy is constrained?

3 Precision Control: Provide training curves using strictly FP16/FP32 to definitively isolate numerical truncation from algorithmic hacking.

4 Strict On-Policy Control: Show the discrepancy curve when using only 1 update step per batch to rule out off-policy drift.

**Limitations:**

Address Experimental Limitations: The authors must explicitly state the limitations of their empirical setup. Specifically, they need to acknowledge:

The limitation of evaluating their hypothesis on a single model architecture (Qwen3-30B-A3B) and a single, arguably saturated benchmark (AIME24), which restricts the generalizability of their claims.

The methodological limitation of explicitly omitting the KL divergence penalty, which fundamentally alters the standard dynamics of RL fine-tuning.

The confounding factors introduced by their specific hyperparameter choices, such as using 4 parameter update steps per batch (introducing off-policy drift) and utilizing BF16 precision without an FP16/FP32 control group to isolate numerical truncation artifacts.

**Strengths And Weaknesses:**

Strengths

1 Novel and Thought-Provoking Perspective: While the community is heavily focused on "Reward Hacking," this paper provides a refreshing pivot by identifying vulnerabilities at the optimization objective level. Framing implementation discrepancies as "Objective-level hacking" is an elegant and insightful theoretical contribution.

2 Elegant Ablation Design: The proactive noise injection experiments—specifically separating biased distortion from variance-based distortion—are cleverly designed. This effectively isolates confounding variables and empirically supports the hypothesis that biased signals drive the collapse.

3 Practical Engineering Value: The paper provides a strong cautionary tale against token-level clipping in MoE RLVR training, highlighting its unintended "double-edged sword" effects. The indirect validation of sequence-level clipping offers actionable advice for practitioners.

Weakness

1 Lack of Model and Task Diversity: The empirical claims rest entirely on a single model (Qwen3-30B-MoE) evaluated on an outdated benchmark (AIME24). For a 2026 conference, AIME24 is saturated for 30B models. The omission of cutting-edge, high-complexity benchmarks (e.g., AIME25, HMMT25) leaves it unclear whether the collapse is true "objective-level hacking" or merely trivial degeneration from signal depletion on an easy task.

2 Omission of the KL Penalty: The authors explicitly remove the KL divergence term, which acts as the critical "brakes" in PPO/GRPO to prevent pathological policy divergence. Without this anchor, the model is practically guaranteed to hack any minor bias, making the "irreversible collapse" unconvincing for standard, safe RLVR setups.

3 Missing Precision Controls: The paper traces the $\pi_{infer}$ clustering to BF16 truncation errors but fails to run an end-to-end FP16/FP32 control. If higher precision eliminates the discrepancy, this is a simple engineering float-point bug, not a profound optimization theory requiring the $\Delta\mathcal{J}(\theta)$ framework.Confounding Off-Policy Effects: Using 4 parameter update steps per batch introduces significant off-policy drift. The authors fail to isolate whether the discrepancy explosion is actually caused by the proposed hacking or simply standard policy degradation from a high Update-To-Data (UTD) ratio.

4 Unclear Impact Scale: The "Clip ratios" in Table 1 are microscopic (e.g., 1.50×10⁻³). The paper lacks statistical justification for how clipping a fraction of a percent of tokens can mechanically trigger a systemic cascade and collapse of a 30B model.

---

> ### Author Rebuttal · Authors · 2026-03-30
>
> Thank you for recognizing the theoretical contribution and the practical engineering value of our work. We have added additional empirical analysis, which we hope could address your questions.
>
> > Weakness1 & Question1 (benchmark stress-test)
>
> We appreciate your suggestion and have further added tests on **AIME25** and **HMMT25 Feb** (under the same setup as in Fig. 2). As shown, **the training-inference mismatch exhibits a generally negative correlation with performance scores**, suggesting that the observed trend is not merely an artifact of signal saturation.
>
> Additionally, **we validate the generality of objective-level hacking across broader datasets and model architectures (Qwen3-4B, Qwen3-8B, DeepSeek-R1-Distill-Qwen-7B)**; due to space limitations,please refer to Tables 1-3 of Reviewer b6pA.
>
> **Table.1** AIME25 & HMMT25Feb Evaluation for Fig.2
> |Gradient Step|200|400|600|800|
> |:-- |--:|--:|--:|--:|
> |AIME25 (TIS)|**50.4**|49.7|**51.1**|**52.0**|
> |AIME25 (no TIS)|49.8|**50.4**|50.2|49.1|
> |HMMT25 (TIS)|**31.5**|**30.8**|**32.2**|**32.8**|
> |HMMT25 (no TIS)|31.1|30.7|31.6|30.8|
>
> > Weakness2 & Question2 (KL-divergence test)
>
> We have added an experiment incorporating KL divergence term into the injected distortion setting of Qwen3-30B-A3B. As shown in Table 2, **KL divergence** (using a typical coefficient of 0.001; higher values could hinder performance) **fails to effectively prevent mismatch growth**.
>
> Our framework offers an explanation for this phenomenon: **mismatch growth is not a passive or incidental effect, but an intrinsically driven process** (see Fig. 6 for details). Regularization alone is often insufficient to constrain it.
>
> **Table.2** Qwen3-30B-A3B (Seq. / injection $\delta=2$ / KL_coeff=0.001)
> |Gradient Step|60|120|180|240|300|360|
> |:-- |--:|--:|--:|--:|--:|--:|
> |Mismatch $[10^{-2}]$|1.34|1.41|1.71|**2.02**|1.51|0.52|
> |KL-loss $[10^{-3}]$|3.29|5.61|14.6|25.5|48.7|**367**|
> |Valid score|60.8|61.7|64.9|67.0|68.4|**38.0**|
>
> > Weakness3 & Question4 (on-policy setting)
>
> We appreciate your comments. To better illustrate this issue, We add an **on-policy experiment on Qwen3-30B-A3B**. Results show mismatch still grows spontaneously, **ruling out off-policy drift as the only cause**.
>
> **Table.3** Qwen3-30B-A3B (Tok. / on-policy)
> |Step|25|50|75|100|125|150|
> |:-- |--:|--:|--:|--:|--:|--:|
> |Mismatch $[10^{-2}]$|1.41|1.43|1.43|1.52|1.58|1.63|
> |Valid score|60.9|61.0|61.8|60.6|61.9|63.9|
>
> Additionally, we would like to clarify that one of the core contribution of objective-level hacking framework is to provide **a unified perspective on multiple factors driving abnormal mismatch growth**, rather than being limited to off-policy effects; we hope this experiment addresses your concern.
>
> > Weakness3 & Question3 (clustering effect and precision)
>
> We thank you for this question, and would like to clarify a possible misunderstanding first: our view is that **mismatch is not merely a numerical precision issue, but arises from intrinsic algorithmic mechanisms**. While we conclude that the probability clustering effect results from limited BF precision, it accounts for **only a small part of mismatch** rather than being its primary cause. We quantify the unbiasedness (Mean) and variance (Std) of such clustering effect in Table 4. Even in extreme cases (details see Appendix C.1), it does not cause significant distribution drift.
>
> **Table.4** Ratio of token probility between FP32 and BF16
> ||Mean|Std|Max|Min|KL Divergence|
> |:-- |--:|--:|--:|--:|--:|
> |Average value|1.0000|0.0048|1.0189|0.9815|0.02422|
>
> From another point, increasing precision to FP32 can mitigate mismatch to some extent; however, **the field is moving toward lower-precision training**, while higher precision offers limited gains at substantial cost. In this context, we hope to further highlight the practical value of the objective-level hacking framework.
>
> > Weakness 4 (Clip ratios)
>
> Thanks for your question. Although the clip ratios in RLVR are indeed small, prior work [2,3] suggests that even a small subset of low-probability tokens can have a significant impact on optimization, since they are usually decision points in reasoning steps. We will provide further discussion in the revised manuscript.
>
> Thanks for your questions and suggestions! We hope this addresses your questions, and would be glad to hear any further suggestions. Thank you!
>
> [1] F. Yao, et. al. Your Efficient RL Framework Secretly Brings You Off-Policy RL Training (2025)
>
> [2] S. Wang, et. al. Beyond the 80/20 Rule: High-Entropy Minority Tokens Drive Effective Reinforcement Learning for LLM Reasoning, NeurIPS (2025)
>
> [3] H. Meng et al. Sparse but Critical: A Token-Level Analysis of Distributional Shifts in RLVR Fine-Tuning of LLMs, ICLR (2026)

---

> > ### Author Rebuttal · Reviewer_yjSN · 2026-04-04
> >
> > I thank the authors for their extensive rebuttal, particularly the addition of Dense model baselines and modern benchmarks (HMMT25). However, a careful review of this new data confirms, rather than alleviates, my core concerns regarding the paper’s soundness and significance.
> >
> > Specifically, three critical issues remain unresolved:
> >
> > 1. Severe Over-claiming and Lack of Universality (W1 & Q1)
> > Table 1 reveals that the mismatch in the Dense model merely fluctuates from 1.059 to 1.081 after 480 steps, completely lacking the claimed exponential explosion. This explicitly proves that "objective-level hacking" is not a universal fundamental law of RLVR optimization, but highly likely an idiosyncratic vulnerability specific to MoE routing mechanisms. Framing an MoE-specific architectural vulnerability as a universal foundation model pathology is a severe over-claim.
> >
> > 2. Exaggerated Rhetoric vs. Actual Data (Q3)
> > The main text heavily relies on catastrophic phrasing like "irreversible collapse" and "irreversible degradation." Yet, the new HMMT25 downstream evaluation (Table 3) shows only a minor ~2-point performance dip (32.8 down to 30.8) without TIS correction. A minor performance degradation on a difficult benchmark is fundamentally different from a systemic "model collapse." This severe disconnect makes the paper's central narrative misleading.
> >
> > 3. Lack of Substantive Mitigation & Significance (W3)
> > The authors concede they offer no novel algorithmic solution, instead defending the framework as an "early-warning indicator." However, monitoring distribution mismatch and KL drift is already standard operational practice in modern RLVR pipelines to detect policy drift. Using a dense theoretical apparatus merely to conclude that practitioners should fall back to an existing heuristic (sequence-level clipping) severely limits the paper's practical significance and contribution for a venue of this caliber.

---

> > > ### Author Response · Authors · 2026-04-05
> > >
> > > We thank the reviewer for the follow-up. To clarify potential misunderstandings, we provide additional discussion and experimental evidence in hope of addressing the concerns raised.
> > >
> > > ## Q1
> > >
> > > Regarding the reviewer's concerns about the empirical analysis on dense models, we provide a point-by-point response below.
> > >
> > > > The claimed exponential explosion of mismatch
> > >
> > > We would like to clarify that **neither in the manuscript nor the rebuttal did we claim an "exponential explosion" of mismatch** for dense models; in fact, we did not use such terminology even for MoE. Our consistent claim is that mismatch can **grow abnormally / spontaneously** during training.
> > >
> > > > Not a universal fundamental law
> > >
> > > We respectfully disagree that the dense model results undermine generality. **Mismatch levels differ between dense and MoE models, but our framework remains fully applicable in both cases, as it continues to characterize the overall mismatch patterns well.** This instead further supports the generality of our framework and is also acknowledged by other reviewers. We further provide detailed training dynamics of Deepseek-R1-Distill-Qwen-7B via an anonymous link in the hope of helping clarify this misunderstanding: <https://anonymous.4open.science/r/Dynamics-deepseek-R1-Distill-Qwen-7B/README.md>
> > >
> > > > Specific to MoE routing mechanisms
> > >
> > > We emphasize that our framework is not limited to MoE models; as discussed above, it applies more generally. We focus on MoE because training instability in MoE models is more pronounced. Besides, we do not agree that works focusing on MoE implies a limited scope. Rather, given that state-of-the-art models predominantly adopt MoE architectures, we believe such studies are well aligned with ongoing research trends and may offer useful insights for future developments.
> > >
> > > ## Q2
> > >
> > > Regarding the claim that terms such as "irreversible collapse" and "irreversible degradation" are "exaggerated rhetoric," we respectfully disagree and believe this wording is appropriate. We clarify as follows:
> > >
> > > (1) The phenomenon of irreversible training collapse in MoE models has been widely reported, and the term "irreversible collapse" is commonly used [1-3]. For example, [1] from the Qwen team states:
> > >
> > > > However, current state-of-the-art RL algorithms exhibit severe stability issues when training gigantic language models, often resulting in catastrophic and irreversible model collapse (Page 1)
> > >
> > > Therefore, we do not believe this wording is misleading.
> > >
> > > (2) Our paper consistently emphasizes **in long-horizon training**, once mismatch grows larger, it leads to significant degradation in capability. The observed drop here is not a fluctuation, but the onset of continued decline, i.e., an early signal of training collapse. We extend this experiment to a longer training horizon in Table 1, where continued training leads to sustained performance degradation instead of improvements (see also [2]). We will include these results into the revised version to clarify similar concerns.
> > >
> > > **Table.1** Token-level clipping in Fig2
> > > |Gradient Step|200|400|600|800|1000|1200|1400|1600|
> > > |:-- |--:|--:|--:|--:|--:|--:|--:|--:|
> > > |Valid|64.3|65.7|65.5|64.7|63.5|**54.6**|**55.3**|**51.6**|
> > >
> > > ## Q3
> > >
> > > **We respectfully disagree with the reviewer’s characterization of our contribution as merely an "early-warning indicator," and would like to reiterate the main conclusions and contributions of our work**:
> > >
> > > (1) **We construct a novel theoretical framework that characterizes the mechanism underlying training instability in RLVR, offering a unified understanding of diverse factors contributing to instability.** To the best of our knowledge, this is the first work to provide a mechanistic account of the abnormal mismatch growth phenomenon, which is a dynamic strongly associated with training instability.
> > >
> > > (2) **We design a set of causal intervention experiments that provide an effective strategy for tracing the root causes of instability.** This paradigm provides a principled reference for future research.
> > >
> > > (3) **Our work delivers practical insights for addressing instability in MoE models.** We generalize prior focus on token-level clipping to broader token-level interventions, showing that operations such as token-level reweighting can also induce instability and should be treated with caution. In contrast, sequence-level interventions offer safer alternatives. These findings provide concrete guidance for engineering practice and offer insights for future stable algorithm design.
> > >
> > > We sincerely thank the reviewer for raising these additional concerns. We appreciate your feedback and hope that our clarifications could help address your concerns. Thanks!
> > >
> > > [1] C. Zheng et al. Group Sequence Policy Optimization (2025)
> > >
> > > [2] J. Liu et al. When Speed Kills Stability: Demystifying RL Collapse from the Training-Inference Mismatch (2025)
> > >
> > > [3] Minimax, MiniMax-M1: Scaling Test-Time Compute Efficiently with Lightning Attention (2025)

---

### Official Review · Reviewer_xurq · 2026-03-12

**Soundness:** 2
**Presentation:** 3
**Significance:** 2
**Originality:** 3
**Overall Recommendation:** 4
**Confidence:** 4

**Summary:**

This paper studies training instability in Reinforcement Learning with Verifiable Rewards (RLVR), with a particular focus on large language models with Mixture-of-Experts (MoE) architectures. The paper argues that a key failure mode is an abnormal and persistent growth of the training-inference discrepancy, and introduces a theoretical framework based on "objective-level hacking" to explain this effect. Unlike standard reward hacking, the proposed phenomenon is attributed to token-level credit misalignment, which induces spurious optimization signals at the objective level.

 To support this claim, the authors analyze the role of token-level weighting and construct controlled interventions on a Qwen3-30B-A3B model. In particular, they inject biased token-level weight distortions by assigning different credit to high-probability and low-probability tokens, and compare this against an ablation that adds unbiased Gaussian variance noise across tokens. The paper argues that biased distortion, rather than variance alone, is the main cause of discrepancy growth. The authors also propose an observable metric based on Equation 20 to monitor objective-level hacking during optimization.

**Compliance With Llm Reviewing Policy:**

Affirmed.

**Final Justification:**

This paper addresses an important and timely problem—training instability in RLVR, particularly for MoE architectures—and introduces a useful conceptual framework ("objective-level hacking") that offers a more structured explanation than purely empirical diagnosis. The distinction between token-level credit misalignment and standard reward hacking is original and well-motivated (Originality: good). The experimental design, particularly the contrast between biased distortion and unbiased Gaussian noise, is a sensible ablation strategy (Soundness: originally fair, now improved).

My initial assessment was a Weak Reject (3), primarily due to: (1) narrow empirical scope concentrated on a single model family, (2) reliance on GSPO as a stable reference without broader validation, (3) the lack of a concrete stabilization method derived from the theory, and (4) minor presentation issues.

The rebuttal has substantively addressed concerns (1), (2), and (4):

- The authors extended experiments to Qwen3-4B, Qwen3-8B, and DeepSeek-R1-Distill-Qwen-7B across both dense and MoE architectures, with an additional training dataset (skywork-or1). The consistent pattern across these settings—including the informative observation that dense models exhibit much milder mismatch than MoE models—meaningfully strengthens external validity.
- Sequence-level clipping results remain consistent across the new settings, reducing the concern that the conclusions are overly tied to a specific GSPO configuration.
- Downstream benchmark results on AIME25 and HMMT25 provide more direct evidence linking mismatch growth to performance degradation, though the margins remain modest and would benefit from further validation.
- The Figure 2 correction is acknowledged and appreciated.

Concern (3) remains: the paper is diagnostic rather than algorithmic, and no new stabilization method is proposed. The authors argue the framework's value lies in guiding practical iteration and providing early-warning indicators, which I find reasonable for an analysis-oriented contribution—but this does limit the immediate practical impact.

**Weighing the dimensions after rebuttal:**
- **Soundness**: Improved from fair to good. The broader experimental coverage addresses the main empirical gap.
- **Originality**: Remains good. The objective-level hacking framework is a novel and clearly articulated conceptual contribution.
- **Significance**: Improved from fair to fair-to-good. The problem is practically important, and the extended evidence makes the analysis more actionable, though the lack of a derived mitigation method limits the significance ceiling.
- **Clarity**: Remains good, pending the promised figure and notation fixes.

Overall, the rebuttal has shifted my assessment. The strengthened empirical foundation, combined with the original conceptual contribution, now tips the balance in favor of acceptance. I raise my score from 3 to 4.

**Key Questions For Authors:**

Q1. How general is the proposed explanation across RLVR settings?

 The current experiments are concentrated on a fairly narrow setup. Could the authors provide additional evidence, or at least a stronger argument, that the objective-level hacking explanation remains valid across more datasets, model families, and RLVR training pipelines?

 Q2. How tightly is the claim tied to GSPO as a stable reference point?

 Several conclusions appear to rely on the observation that sequence-level clipping does not exhibit the same discrepancy-growth pathology in the current setting. Do the authors have evidence that this remains true more broadly, or are there regimes where GSPO can also fail in similar ways?

 Q3. Can the authors clarify how training-inference discrepancy relates to downstream generalization?

 The paper argues that discrepancy growth is a meaningful instability signal. It would be helpful to understand more directly how this metric correlates with model quality, generalization, or failure behavior beyond the optimization traces themselves.

 Q4. Can the authors revise Figure 2 for clarity?

 Figure 2 currently seems to have caption-to-plot color mismatches, and abbreviations such as `Tok` are not clearly defined in a self-contained way. Cleaning this up would make the paper easier to read and evaluate.

**Limitations:**

yes

**Strengths And Weaknesses:**

Strengths:

 S1 (Originality / soundness). The paper proposes a clear and interesting conceptual distinction between standard reward hacking and the paper's notion of objective-level hacking. This gives the instability phenomenon a more explicit theoretical interpretation than a purely empirical diagnosis.

 S2 (Soundness). The empirical design is reasonably targeted. The contrast between biased token-level distortion and unbiased Gaussian noise is a sensible way to isolate whether the instability is driven by structured credit misalignment rather than variance alone.

 S3 (Significance). Training instability in RLVR is an important problem, especially for large-scale MoE systems where optimization pathologies can be costly and hard to debug. A better mechanistic understanding of discrepancy growth could be useful to the community even if the paper is more diagnostic than algorithmic.

 Weaknesses:

 W1 (Soundness). The empirical scope is narrow. Most of the evidence is concentrated on a single model family and a relatively limited training setting, which makes it difficult to assess how broadly the proposed explanation applies across architectures, datasets, and RLVR pipelines.

 W2 (Soundness / significance). Some of the central conclusions rely on the premise that sequence-level clipping methods such as GSPO are intrinsically more stable and do not exhibit the same anomalous discrepancy growth. While the paper presents evidence for this in the studied setup, it is not yet clear whether this conclusion is robust across more diverse conditions.

 W3 (Significance / originality). The paper is stronger at problem formulation and diagnosis than at mitigation. Although it discusses existing approaches such as TIS and GSPO, it does not derive a new stabilization method directly from the proposed theory. As a result, the practical takeaway is more limited than the conceptual framing.

 W4 (Presentation). Some presentation details need cleanup. For example, Figure 2 appears to contain notation and color-caption mismatches that make the figure harder to parse than necessary. This is fixable, but it weakens clarity in an otherwise technical paper.

---

> ### Author Rebuttal · Authors · 2026-03-30
>
> Thanks for your appreciation of the significance and theoretical contribution of our work. We have added additional empirical experiments, which we hope could help address your questions.
>
> > Weakness 1 & Question 1 (validation across more scenarios)
>
> We appreciate your valuable suggestion and **have extended the empirical analysis across different model architectures (Qwen3-4B, Qwen3-8B), cold-start initializations (DeepSeek-R1-Distill-Qwen-7B), and training datasets (skywork-or1)**. We present the mismatch dynamics for DeepSeek-R1-Distill-Qwen-7B experiments in Table 1; due to space limitations, results for Qwen3-4B and Qwen3-8B are provided in Tables 2,3 of Reviewer b6pA.
>
> **The overall pattern remains well explained by the objective-level hacking framework**. For example, dense models also exhibit spontaneous mismatch growth, which can be mitigated by TIS or sequence-level clipping; strong credit misalignment injection triggers mismatch and collapse. Notably, Mismatch growth in dense models is much less severe than in MoE models (1.059->1.081 vs. 1.418->2.664 at the same gradient steps).
>
> In addition, we conduct experiments under diverse MoE settings: for example, KL divergence regularization fails to prevent mismatch growth (see Table 2 of Reviewer yjSN). **Our experiments show that objective-level hacking generalizes across diverse model architectures and scenarios, not limited to Qwen3 routing design and specific settings.**
>
> **Table.1** Mismatch [$10^{-2}$] on Deepseek-R1-Distill-Qwen-7B
> |Gradient Step|60|120|180|240|300|360|420|480|Valid score|
> |:-- |--:|--:|--:|--:|--:|--:|--:|--:|--:|
> |Tok. |1.059|1.055|1.054|1.071|1.064|**1.076**|**1.077**|**1.081**|46.3|
> |Tok.+TIS|1.058|1.049|1.064|1.067|1.062|1.056|1.048|1.047|47.1|
> |Seq.|1.054|1.049|1.048|1.061|1.062|1.064|1.060|1.047|47.3|
> |Seq.+($\delta=3$)|1.055|1.055|1.061|**1.124**|-|-|-|-|collapse|
>
> > Weakness 2 & Question 2 (sequence-level clipping)
>
> **We have extended sequence-level clipping experiments on DeepSeek-R1-Distill-Qwen-7B (Table 1) and Qwen3-4B (Table 2), showing that pure sequence-level clipping does not induce significant anomalous mismatch growth under these settings**. It is also worth noting that mismatch growth in dense models is much less severe than in MoE, and we do not observe significant gains from GSPO for dense models.
>
> **Table.2** Mismatch [$10^{-2}$] on Qwen3-4B
> |Gradient Step|60|120|180|240|300|360|420|480|Valid Score|
> |:-- |--:|--:|--:|--:|--:|--:|--:|--:|--:|
> |Tok.|1.004|0.999|0.999|1.004|1.005|**1.013**|**1.026**|**1.062**|47.9|
> |Seq.|0.999|0.990|0.968|0.974|0.984|0.969|0.960|0.987|48.1|
>
> > Weakness 3 (practical engineering value)
>
> We appreciate your comments and would like to further supplement the practical values of our framework. Our framework focuses on the training collapse in MoE models, which has been widely reported by the community [1,2]. However, the mechanisms underlying collapse dynamics still lack a clear theoretical explanation. As a result, when collapse occurs (e.g. when empirically motivated algorithmic modifications trigger unexpectedly severe instability), there is often no clear direction for intervention.
>
> In this context, objective-level hacking framework systematically uncovers the mechanisms behind collapse through extensive experiments. While it does not provide a standalone recipe, **it enables faster practical iteration by clarifying what to avoid and why**, thereby reducing resource waste. Moreover, as an interpretability-oriented framework, it has the potential to inspire applications. For example, it provides an early-warning indicator for abnormal training dynamics (Appendix D.2), allowing issues to be identified earlier.
>
> > Question 3 (downstream generalization)
>
> We understand your concern and have included additional benchmark tests (AIME25, HMMT25) in Table 3 to better demonstrate that **mismatch generally shows a negative correlation with performance scores**. The negative impact of mismatch on downstream task performance has also been reported in prior work [3].
>
> **Table.3** Evaluation for Fig.2
> |Gradient Step|200|400|600|800|
> |:-- |--:|--:|--:|--:|
> |AIME25 (TIS)|**50.4**|49.7|**51.1**|**52.0**|
> |AIME25 (no TIS)|49.8|**50.4**|50.2|49.1|
> |HMMT25 (TIS)|**31.5**|**30.8**|**32.2**|**32.8**|
> |HMMT25 (no TIS)|31.1|30.7|31.6|30.8|
>
> > Weakness 4 & Question 4 (Figure 2)
>
> Thanks for your suggestions. In Fig.2, the correct colors are blue for token-level clipping and red for TIS. We will revise this typo and add definitions for abbreviations (e.g. Tok) in the main text to improve clarity.
>
> Thanks for your constructive suggestions! We sincerely look forward to any further discussions with you.
>
> [1] C. Zheng et al. Group sequence policy optimization (2025)
>
> [2] W. Ma et al. Stabilizing MoE Reinforcement Learning by Aligning Training and Inference Routers (2025)
>
> [3] F. Yao, et al. Your Efficient RL Framework Secretly Brings You Off-Policy RL Training (2025)

---

> > ### Author Rebuttal · Reviewer_xurq · 2026-04-03
> >
> > I thank the authors for the substantial additional experiments. My main concerns are largely addressed:
> >
> >   - W1/Q1: The extended experiments across Qwen3-4B, Qwen3-8B, and DeepSeek-R1-Distill-Qwen-7B, with an additional training dataset, meaningfully broaden the empirical scope. The observation that dense models exhibit much milder mismatch than MoE models is an informative finding in itself.
> >   - W2/Q2: The sequence-level clipping results remain consistent across the new settings, which strengthens the claim.
> >   - Q3: The AIME25/HMMT25 results provide more direct evidence linking mismatch to downstream performance, though the margins are modest.
> >   - W4/Q4: The figure correction is appreciated.
> >
> >   One remaining concern is W3: the paper remains diagnostic rather than algorithmic. The authors acknowledge this and argue the framework's value lies in guiding practical iteration and early-warning monitoring. While I find this reasonable for an analysis-oriented paper, the practical impact would be stronger with a concrete stabilization method derived from the theory.
> >
> >   Overall, the rebuttal has substantially strengthened the empirical foundation. I am willing to raise my score to reflect the improved evidence, while noting that the contribution remains primarily analytical.

---

> > > ### Author Response · Authors · 2026-04-03
> > >
> > > Thank you for your valuable feedback and for supporting our work. We are glad to hear that the additional experiments help strengthen the paper. In the revised manuscript, we will include the empirical analysis on dense models and further discuss the more severe training instability of MoE models to better clarify our motivation.
> > >
> > > We sincerely appreciate your insightful question regarding the analysis-oriented nature of our framework, as well as your suggestion that a concrete stabilization method would further strengthen the practical impact. This is an important and inspiring direction, and well worth future investigation. Within our current study, which is primarily oriented toward interpretability, we do not yet provide a concrete stabilization method; instead, our contributions in terms of practical takeaways are primarily centered on a set of flexible stabilization insights or a principled and general-purpose strategy.
> > >
> > > For example, one key takeaway from the objective-level hacking framework is that, when instability arises, it is preferable to prioritize sequence-level interventions over token-level ones, since heuristic token-level interventions tend to introduce training instability, especially in long-horizon training. Sequence-level interventions are not only limited to sequence-level clipping, but also include a broader range of operations such as sequence-level rollout sample filtering. For example, one can filter out degenerate samples based on repetitive patterns, such as long-span token repetition. We will include more detailed discussions on these sequence-level intervention tips in the revised manuscript, with the aim of providing richer practical insights beyond the interpretability-focused perspective.
> > >
> > > Thanks for this valuable and inspiring suggestion. We will incorporate a more detailed discussion of these in both the main text and the limitation part, outlining broader perspectives beyond the interpretability focus as well as potential avenues for extending the framework, with the hope of inspiring future work in this direction. Thank you again for helping us improve the quality of our work!

---

### Official Review · Reviewer_2tF4 · 2026-03-13

**Soundness:** 3
**Presentation:** 3
**Significance:** 3
**Originality:** 3
**Overall Recommendation:** 4
**Confidence:** 2

**Summary:**

In this paper, authors introduces a framework for understanding training instability in reinforcement learning RL, particularly in  MoE architectures. They propose the novel concept of "objective-level hacking", where RL inadvertently trains on a different objective than the ideal / intended one. They further claim that object-level hacking arises from token-level credit misalignment. The authors formalize how this mechanism drives the abnormal growth of the training-inference discrepancy, a key pathological dynamic in MoE training.  Experiments on a 30B MoE model using the Qwen3-30B-A3B architecture empirically shows that stronger token-level clipping and train-inference shift accelerates discrepancy growth.

**Compliance With Llm Reviewing Policy:**

Affirmed.

**Final Justification:**

I have read other reviewer's comments and I share concerns raised by yjSN. However, I still think this paper has an original framing to understand RLVR training instability. I will maintain my score but updated my confidence.

**Key Questions For Authors:**

1. The theoretical framework and analysis presented in this work appear to be general and not specific to MoE architectures. If so, why are all experiments conducted exclusively on a 30B MoE model? While the authors acknowledge that MoE models may be particularly sensitive to these instabilities, I would still expect qualitatively similar behaviors to manifest in dense models, albeit potentially less prominently. Have the authors conducted any experiments on dense model architectures to verify this, and if so, how do the resulting training dynamics compare?

**Limitations:**

yes

**Strengths And Weaknesses:**

**Strengths**:

1. Authors introduces a novel conceptual framework "objective-level hacking"  to explain training instabilities in MoE-based RLVR. This framework is an original and thought-provoking perspective. I can see this paper opens the door to deeper theoretical investigations into the root causes of instability.

2. The empirical evaluation is thorough and well-designed. The experiments examining different clipping strengths and intervention strategies such as TIS are comprehensive. I also liked the causal injection experiments provide further evidence that token weight misalignment is not just correlational but causal to training instabilities.

**Weakness**:

1. One of the central claims of the paper is a causal relationship between objective-level hacking and the training-inference discrepancy. However, I can see some qualitative differences inconsistency between Figure 4 and Figure 5: while Figure 4 shows a gradual and cumulative drift in the discrepancy, the noise injection experiments in Figure 5 produce a sudden and catastrophic performance collapse. This qualitative difference in dynamics raises questions about whether the injected distortion faithfully reproduces the naturally occurring instability mechanism.

2. While the paper provides a unified perspective on the root causes of RLVR training instability, authors did not attempt to use this framework to explain how existing mitigation strategies reduce objective-level hacking. Incorporating such an analysis into the proposed framework might make the paper's findings more comprehensive.

---

> ### Author Rebuttal · Authors · 2026-03-30
>
> Thank you for appreciating the novelty of our framework and its role as a pioneering effort toward understanding the root causes of instability. We have added additional analyses and experiments in hope of addressing your questions.
>
> > Weakness 1 (Instability mechanism)
>
> We appreciate this insightful question and would like to further clarify the analysis and motivation of the injected distortion experiments, aiming to better present our logical chain.
>
> We would like to note that **the collapse dynamics in the injection experiments are highly sensitive to the amplitude**. We summarize in Table 1 the steps at which model performance drops significantly, along with the corresponding mismatch, showing that weaker injections lead to slower mismatch growth and a later, more gradual degradation. The abruptness of the collapse is not an inherent property of the injection pattern, but is also determined by the amplitude.
>
> **Table.1** Collapse dynamics (Fig. 5)
> |Injection amplitude|Collapse step|Mismatch [$10^{-2}$]|
> |:-- |--:|--:|
> |$\delta=3$|260|3.11|
> |$\delta=2$|320|2.45|
> |$\delta=1.2$|680|1.58|
>
> Additionally, the injected distortion experiment indeed simplifies the mechanism of credit misassignment compared to natural settings (clipping or initial discrepancy), by only distinguishing between high- and low-probability tokens; but **it isolates the core mechanism**: they share the same core similarity, token-level credit misalignment and both lead to abnormal mismatch growth, thereby providing evidence for our conclusions.
>
> > Weakness 2 (discussion on existing strategies)
>
> Thanks for this constructive suggestion. We have provided an interpretation of the effectiveness of existing mitigation strategies under the objective-level hacking framework.
>
> (**Data filtering**) Filtering low-quality samples (e.g. repetitive rollouts) is a common stabilization practice. In the objective-level hacking framework, such methods do not introduce sereve MoE instability because **they inherently operate at the sequence level**, which is safer for MoE models.
>
> (**Algorithms**) Algorithms such as truncated importance sampling or GSPO do not reduce mismatch at the infrastructure level, but instead **mitigate or break the objective-level hacking effect at the objective design level**, thereby maintaining training stability.
>
> (**Infrastructure**) Methods such as routing replay can **reduce the initial mismatch**, which also mitigates the objective-level hacking effect (see Fig. 8) and helps avoid early entering a positive feedback loop during training.
>
> A more detailed discussion will be included in the revised manuscript.
>
> > Question 1 (Validation on dense models)
>
> We appreciate this valuable suggestion, which is important for improving the generality of our framework. **We have extended our empirical analysis to dense models with different architectures (Qwen3-4B, Qwen3-8B) and cold-start initialization (DeepSeek-R1-Distill-Qwen-7B), using additional training data (skywork-or1)**. The mismatch dynamics are reported in Table 2; due to space limitations, results for Qwen3-4B and Qwen3-8B are provided in Tables 2,3 of Reviewer b6pA.
>
> **Table.2** Mismatch [$10^{-2}$] on Deepseek-R1-Distill-Qwen-7B
> |Gradient Step|60|120|180|240|300|360|420|480|Valid score|
> |:-- |--:|--:|--:|--:|--:|--:|--:|--:|--:|
> |Tok. |1.059|1.055|1.054|1.071|1.064|**1.076**|**1.077**|**1.081**|46.3|
> |Tok.+TIS|1.058|1.049|1.064|1.067|1.062|1.056|1.048|1.047|47.1|
> |Seq.|1.054|1.049|1.048|1.061|1.062|1.064|1.060|1.047|47.3|
> |Seq.+($\delta=3$)|1.055|1.055|1.061|**1.124**|-|-|-|-|collapse|
>
> The dynamics show that **dense model follows the pattern described by the objective-level hacking**: they also exhibit anomalous mismatch growth, which can be mitigated by methods such as TIS or sequence-level clipping; strong credit misalignment injection leads to increased mismatch and model collapse. The abnormal mismatch behavior in dense models has also been reported by the community [1,2].
>
> **It is worth noting that mismatch growth in dense models is far less severe than in MoE models.**.  Dense models show only a slight mismatch increase with a lower initial value (1.059->1.081, see Table.1), while MoE models rise much more at the same gradient steps (1.418->2.664). Additional mismatch metrics, Pearson correlation coefficient, also confirm this (Dense: 0.9996->0.9995; MoE:0.9982->0.9804). The major driver for the severe instability in MoE models is **expert reactivation**: different experts activated in training vs. inference create more room for mismatch growth [3].
>
> Overall, thank you for your valuable suggestions! We sincerely look forward to further discussions. Thanks!
>
> [1] F. Yao, et al. Your Efficient RL Framework Secretly Brings You Off-Policy RL Training (2025)
>
> [2] He, Horace and Thinking Machines Lab, Defeating Nondeterminism in LLM Inference (2025)
>
> [3] W. Ma et al. Stabilizing MoE Reinforcement Learning by Aligning Training and Inference Routers (2025)

---

> > ### Author Rebuttal · Reviewer_2tF4 · 2026-04-03
> >
> > Thank you for the detailed response! I really liked your comparison between dense and MoE models and I think the discussion on implication to mitigation strategy helped me get some intuitions behind why they work and how they work. I would like to keep my score.

---

> > > ### Author Response · Authors · 2026-04-03
> > >
> > > Thank you very much for your thoughtful and encouraging feedback. We are very glad that our empirical analysis comparing dense and MoE models, together with the discussion of existing mitigation strategies within our framework, helped clarify the underlying intuitions.
> > >
> > > We sincerely appreciate your positive assessment and your recognition that the concerns have been adequately addressed. We will ensure that these related discussions and results are incorporated into the revised manuscript, and hope that the additional clarifications further improve the quality of our work. Thank you again for your time and for the valuable feedback you have provided on our work!

---

### Official Review · Reviewer_b6pA · 2026-03-13

**Soundness:** 3
**Presentation:** 3
**Significance:** 2
**Originality:** 2
**Overall Recommendation:** 4
**Confidence:** 2

**Summary:**

This paper introduces a framework to understand RLVR training instability in MoE models through the lens of objective-level hacking. The authors show that both the initial training-inference discrepancy and token-level clipping act as sources of objective-level hacking, and that biased rather than variance-based distortion is the primary driver. Experiments on a 30B MoE model support these claims.

**Compliance With Llm Reviewing Policy:**

Affirmed.

**Final Justification:**

The rebuttal addressed my main concerns. Additional experiments across dense and MoE architectures validate the framework's generality, and the noise injection ablations support the causal claim. My residual concern on direct validation of the mismatch-collapse threshold on the 30B model stands but is partially mitigated by the AIME25/HMMT25 results.

On the inter-reviewer disagreement: yjSN's MoE-specificity concern is partially legitimate but a bit overstated: the authors demonstrate applicability to dense models and provide a mechanistically coherent explanation for the severity difference via expert reactivation. The "irreversible collapse" critique is a fair presentation note; the extended training curve (valid score 65.7 → 51.6 over 1600 steps) further supports the claim. I maintain my score of 4.

**Key Questions For Authors:**

- The paper focuses on MoE models, but token-level credit misalignment may exist in dense models too. If the same mechanism cause more instability in MoE specifically, is there a theoretical or empirical account of this difference?

- The discrepancy metric is defined as the standard deviation of importance weights at a training step. How sensitive are the conclusions to this choice of metric, and would alternative measures of distribution mismatch tell a different story?

**Limitations:**

The paper has some discussions on limitations at the end, but would benefit from expansion

**Strengths And Weaknesses:**

Strengths:

- The paper has a good balance between theoretical analysis and empirical experiments. The gap identified is timely and important,  interpretability-focused analysis of MoE training instability is indeed a gap, and this work gives a thorough mechanistic account.
- The experiments (showing objective-level hacking from token-level weight distortion drives training-inference discrepancy growth) are well-designed and the bias-vs-variance ablation (Section 5.1) isolates the key mechanism well.

Weaknesses:

- Only one MoE model (Qwen3-30B-A3B) is tested. At minimum, a second MoE architecture is needed to support generalizability claims. It is unclear whether the findings are specific to Qwen3's routing mechanism or apply broadly to MoE models.
- The paper focuses on MoE models, but does not clearly explain why the objective-level hacking mechanism would be specific to MoE rather than dense models. If the same token-level credit misalignment exists in dense models but causes less instability, the paper should explain why.
- The training-inference discrepancy metric (standard deviation of importance weights) is a reasonable proxy, but it is not validated against direct measures of model degradation. Is it possible for the metric to grow without causing meaningful performance collapse?
- Figure 2's caption describes grey and green curves, but the actual figure shows red and blue. There is no legend explanation for what the colors represent.

---

> ### Author Rebuttal · Authors · 2026-03-30
>
> We sincerely thank you for your appreciation of the novelty and significance of our work. For your construtive suggestions, we have added additional empirical experiments in the hope of adequately addressing your concerns.
>
> > Weakness 1,2 & Question 1 (validate on more models)
>
> We appreciate your suggestion to validate across a broader range of models, which is important for strengthening the generality of our framework. **We have extended the empirical analysis to a broader range of models with different architectures (Qwen3-4B, Qwen3-8B) and different cold-start initializations (DeepSeek-R1-Distill-Qwen-7B)**, based on additional training sets (skywork-or1), shown in Table 1-3.
>
> **The overall pattern is still well explained by the objective-level hacking framework.** For example, anomalous discrepancy growth also appears in dense models; TIS or sequence-level clipping can mitigate this growth; strong credit misalignment injection leads to increased mismatch and collapse. Abnormal mismatch behavior in dense models has also been reported by the community [1,2].
>
> **Table.1** Mismatch [$10^{-2}$] on Deepseek-R1-Distill-Qwen-7B
> |Gradient Step|60|120|180|240|300|360|420|480|Valid score|
> |:-- |--:|--:|--:|--:|--:|--:|--:|--:|--:|
> |Tok. |1.059|1.055|1.054|1.071|1.064|**1.076**|**1.077**|**1.081**|46.3|
> |Tok.+TIS|1.058|1.049|1.064|1.067|1.062|1.056|1.048|1.047|47.1|
> |Seq.|1.054|1.049|1.048|1.061|1.062|1.064|1.060|1.047|47.3|
> |Seq.+($\delta=3$)|1.055|1.055|1.061|**1.124**|-|-|-|-|collapse|
>
> **Table.2** Mismatch [$10^{-2}$] on Qwen3-4B
> |Gradient Step|60|120|180|240|300|360|420|480|Valid Score|
> |:-- |--:|--:|--:|--:|--:|--:|--:|--:|--:|
> |Tok.|1.004|0.999|0.999|1.004|1.005|1.013|**1.026**|**1.062**|47.9|
> |Tok.+TIS|1.001|1.004|0.996|1.011|0.985|1.007|1.004|0.976|51.7|
>
> **Table.3** Mismatch [$10^{-2}$] on Qwen3-8B
> |Gradient Step|60|120|180|240|300|360|420|480|520|Valid Score|
> |:-- |--:|--:|--:|--:|--:|--:|--:|--:|--:|--:|
> |Tok.|1.024|1.043|1.016|1.011|1.022|1.027|**1.032**|**1.060**|**1.077**|55.8|
> |Tok.+TIS|1.035|1.037|1.042|1.009|0.986|1.012|1.028|1.016|1.034|55.2|
>
> Notably, Mismatch growth in dense models is much less severe than in MoE models (1.059->1.081 vs. 1.418->2.664 at the same gradient steps). The major driver of the severe instability in MoE models is **expert reactivation**: different experts activated in training vs. inference create more room for mismatch growth [3]. **Our experiments show that objective-level hacking generalizes across diverse model architectures including dense ones. The increased instability in MoE arises from the expert activation mechanism, which is inherent to MoE and not specific to the Qwen3 routing design.**
>
> > Weakness 3 (mismatch grow and performance)
>
> Thank you for raising this point. We provide empirical evidence of **the consistent negative correlation between training-inference mismatch and model performance across multiple benchmarks (AIME25, HMMT25Feb) (see Reviewer yjSN, Table 1)**. The negative impact of mismatch on downstream task performance has also been reported in prior work [1].
>
> For your question, "Is it possible for the metric to grow without causing meaningful performance collapse?" the answer is yes when mismatch is small: in early training, mismatch can grow without immediate performance degradation (Fig. 2). However, once it exceeds a (potential) threshold, performance degrades clearly. The explanation is that hacking effects become significant at high mismatch level, at which point they noticeably disrupt optimization and degrade model capability.
>
> > Question 2 (alternative measures of mismatch)
>
> We understand your concern about metric robustness and have validated the robustness of our metric using an alternative measure **the Pearson correlation coefficient** (grounded in correlation analysis), in contrast to original importance weights (grounded in importance sampling). **Despite their different statistical foundations, both yield consistent results**. As shown in Fig. 2(c) and Appendix Fig. 9(a), increasing importance weights consistently correspond to decreasing Pearson correlation coefficient, both indicating a growing mismatch, which confirms the stability of the importance weight metric.
>
> > Weakness 4 (Figure 2's caption)
>
> Thanks for pointing out this typo. The correct colors are blue for "token-level clipping" and red for "TIS correction." We will fix this in the revised manuscript.
>
> Thank you for the constructive suggestions. We would greatly appreciate any further discussions. Thanks!
>
> [1] F. Yao, et al. Your Efficient RL Framework Secretly Brings You Off-Policy RL Training (2025)
>
> [2] He, Horace and Thinking Machines Lab, Defeating Nondeterminism in LLM Inference (2025)
>
> [3] W. Ma et al. Stabilizing MoE Reinforcement Learning by Aligning Training and Inference Routers (2025)

---

> > ### Author Rebuttal · Reviewer_b6pA · 2026-04-03
> >
> > Thank you for the rebuttal. The additional experiments across dense and MoE architectures address my concerns on generalizability. A pressing issue presented by this paper, the causal link between mismatch growth and instability, is supported by the noise injection ablations. My concerns are fully resolved. I maintain my score, which already reflects a positive view of the work.

---

> > > ### Author Response · Authors · 2026-04-05
> > >
> > > We sincerely appreciate your positive assessment and are pleased that the additional experiments across dense and MoE architectures alleviated your concerns regarding the generalizability of our framework. We also thank you for your recognition of the noise injection ablations in supporting the causal link discussed in our paper.
> > >
> > > We are grateful for your insightful suggestions and will ensure that the above discussions are carefully incorporated into the revised version. Thank you again for the time and effort you have devoted to helping enhance the quality of our work.

---

### Decision · Program_Chairs · 2026-04-30

**Decision:**

Accept (regular)

**Comment:**

This paper studies RLVR instability in MoE models through the lens of "objective-level hacking", arguing that train-inference discrepancy and token-level clipping can distort the effective optimization objective. Three reviewers ended up positive, and I think the paper makes a useful diagnostic contribution. The rebuttal added helpful dense-model experiments, additional analyses of mismatch/performance correlation, and causal/noise-injection evidence that strengthened the empirical case.

The main remaining concern, raised by the negative reviewer, is whether the proposed framework adds explanatory value beyond routing inconsistency and other MoE-specific train-inference mismatch effects. I take this concern seriously, and I do think the final paper should be more careful about causal language and should explicitly acknowledge routing-based explanations as a plausible confound or complementary mechanism. I also agree that the contribution is primarily analytical/diagnostic rather than algorithmic, and the final framing should reflect that.

However, I think the paper has enough value for being accepted. It targets an important practical problem, provides a coherent conceptual lens, and backs it with fairly careful experiments. The final version should substantially narrow too strong claims and more directly situate objective-level hacking relative to routing-based explanations.